# The Geometry of Mixability

**Armando J. Cabrera Pacheco**                                        *a.cabrera@uni-tuebingen.de*
*University of Tübingen and Tübingen AI Center,*
*Germany*

**Robert C. Williamson**                                      *bob.williamson@uni-tuebingen.de*
*University of Tübingen and Tübingen AI Center,*
*Germany*

**Reviewed on OpenReview:** *https://openreview.net/forum?id=VrvGHDSzZ7*

## Abstract

Mixable loss functions are of fundamental importance in the context of prediction with expert advice in the online setting since they characterize fast learning rates. By re-interpreting properness from the point of view of differential geometry, we provide a simple geometric characterization of mixability for the binary and multi-class cases: a proper loss function $\ell$ is $\eta$-mixable if and only if the superprediction set $\mathrm{spr}(\eta\ell)$ of the scaled loss function $\eta\ell$ slides freely inside the superprediction set $\mathrm{spr}(\ell_{\log})$ of the log loss $\ell_{\log}$, under fairly general assumptions on the differentiability of $\ell$. Our approach provides a way to treat some concepts concerning loss functions (like properness) in a "coordinate-free" manner and reconciles previous results obtained for mixable loss functions for the binary and the multi-class cases.

## 1 Introduction

In the context of *prediction with expert advice* as described by Vovk (1998; 2001), an information game is considered between three players: the *learner*, $N \in \mathbb{N}$ *experts* and *nature*[1]. At each step $t \in \mathbb{Z}_+$,

- each expert makes a prediction $\gamma_i^t \in \Gamma$, $i = 1, ..., N$, (here $\Gamma$ is the *prediction space*) which the learner is allowed to see,
- the learner makes a prediction $\gamma^t \in \Gamma$,
- nature chooses an outcome $\omega^t \in \Omega$ (here $\Omega$ is the *set of outcomes*),
- for a fixed *loss function* $\ell \colon \Omega \times \Gamma \longrightarrow [0, \infty]$, the cumulative loss is calculated for the learner and each of the experts:

$$L_t(\text{learner}) := \sum_{i=1}^{t} \ell(\omega^i, \gamma^i),$$

$$L_t(\text{expert}_k) := \sum_{i=1}^{t} \ell(\omega^i, \gamma_k^i),$$

where $\text{expert}_k$ denotes the $k$-th expert $(k = 1, ..., N)$.

The goal is to bound the difference between the learner's loss and the experts' loss, which is often called the *regret*. That is, we want to bound the quantity

$$R_k^t := L_t(\text{learner}) - L_t(\text{expert}_k)$$

for all $k$.

---

[1] We refer the reader to (Vovk, 1998; 2001) for discussions about the desired (or required) assumptions on $\Gamma$, $\Omega$ and $\ell$.

### 1.1 Mixable games and characterizations of mixable and fundamental loss functions

For a wide class of games, called $\eta$-*mixable games* for $\eta > 0$, the *Aggregating algorithm* (see for example (Vovk, 2001)) ensures an optimal bound for the regret ($R_k^t \leq \eta^{-1} \ln N$, for all $k = 1, ..., N$) independent of the trial $t$. Since the mixability of a game depends on the loss function $\ell$, a loss function $\ell$ is $\eta$-*mixable* if the corresponding game is mixable. Since arguably the aggregating algorithm is one of the most well founded and studied prediction algorithms, there is a natural interest in understanding properties and characterizations of mixable loss functions.

Examples of mixable loss functions include the log loss, relative entropy for binary outcomes (Haussler et al., 1998) and the Brier score (Vovk & Zhdanov, 2009; van Erven et al., 2012). Mixability of a loss function $\ell$ is characterized by a "stronger convexity" of the superprediction set of $\ell$, which can be described as the convexity of the superprediction set of $\ell$ after an "exponential projection" (see (1.3) below and Vovk (2015) and van Erven et al. (2012)). Unfortunately, this characterization of mixability lacks a transparent geometric interpretation.

The main goal of this work is to provide such geometric interpretation. The motivation stems from an observation made by Vovk (2015): a $\eta$-mixable loss can be characterized as the positiveness of the infimum of the quotient of the curvatures of the a strictly proper loss function $\ell$ and the log loss $\ell_{\log}$ for binary outcomes. Here as usual, loss functions are defined on the 2-simplex $\Delta^2$ (see (1.1)). Moreover, he then proves that *fundamentality* (see Vovk (Vovk, 2015)) of a loss can be characterized as the finiteness of the supremum of the same quotient of curvatures. These two results suggest that these properties are *geometric*, meaning that they can be studied using differential geometry tools, and in this regard, mixability and fundamentality should not depend on the coordinates chosen to express them.

Loosely speaking, in convex geometry a convex set $L$ is said to *slide freely inside a convex set* $K$, if for any point $x$ in the boundary of $K$, there is a translation vector $y$ such that the translation of $L$ by $y$ (i.e., the Minkowski sum $L + y$, see (4.2)), intersects $K$ at $x$, and $L + y \subset K$. We provide the following geometric characterization of mixability and fundamentality, as a geometric comparison to the log loss (see Figure 1). Let $\mathrm{spr}(\ell)$ denote the superprediction set of a loss function $\ell$ (see (1.4)).

**Theorem 1.1** (Informal statement). *A continuously twice differentiable proper loss function is $\eta$-mixable if and only there is $\eta > 0$ such that $\mathrm{spr}(\eta\ell)$ slides freely inside $\mathrm{spr}(\ell_{\log})$. In addition, the same $\ell$ is fundamental if and only if there exists $\gamma > 0$ such that $\mathrm{spr}(\ell_{\log})$ slides freely inside $\mathrm{spr}(\gamma\ell)$.*

This new characterization of mixability appeals because it is stated directly in terms of the superprediction sets of the loss functions, rather than the exponentiated super-prediction set. In order to obtain this theorem it is necessary to re-interpret properness from a differential geometry point of view, which constitutes a big part of this work. However, this technical effort pays off. van Erven et al. (2012) characterized $\eta$-mixable (differentiable) loss functions for multi-class loss functions and moreover, related $\eta$ to the Hessian of the *Bayes risk* of $\ell$ and the log loss (see Definition 1.3), which is interpreted as its curvature. By generalizing the tools developed here for the binary case, we were able to obtain a multi-class analog result to Theorem 1.1 and to build a bridge to the results in (van Erven et al., 2012)[2].

Finally we note that the large field of information geometry (Amari, 2016) strives to understand the basic problems of statistical inference from a geometric perspective, with a strong focus on the geometry of statistical manifolds. Likewise, there are strongly geometric theories which describe the effect of hypothesis classes in machine learning (Mendelson & Williamson, 2002; Van Erven et al., 2015). Our hope is that by geometrizing the theory of loss functions we may be able to bridge these diverse perspectives.

### 1.2 Description of results and structure of the article

Using the same setting as (van Erven et al., 2012), we obtain a geometric characterization of $\eta$-mixable loss functions in the sense of differential geometry. Loss functions are considered to be maps $\ell: \Delta^n \longrightarrow \mathbb{R}^n_{\geq 0}$, which under the conditions assumed in this work, give rise to submanifolds $\ell(\mathrm{relint}(\Delta^n))$ of $\mathbb{R}^n$ whose geometric

---

[2]In fact, the present paper can be seen as the successful realization of the attempt of (van Erven et al., 2012), where we felt that the whole argument should be doable in a coordinate-free manner, but did not then have the skills to achieve that.

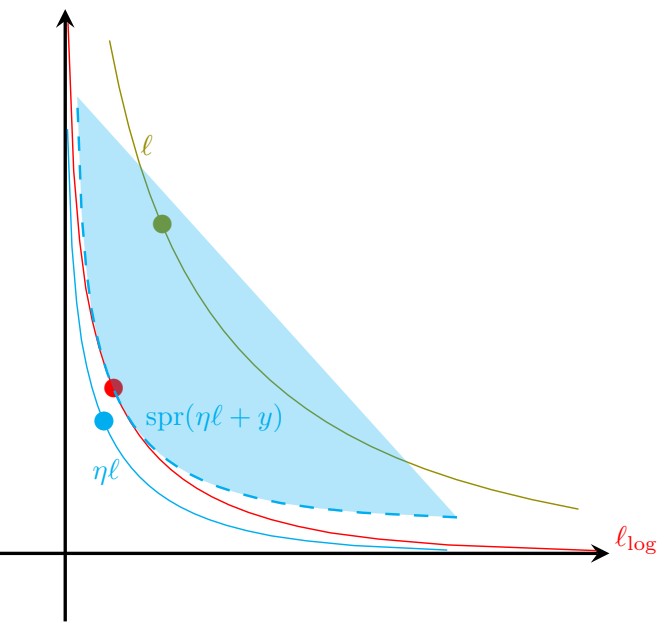

Figure 1: We abuse notation and denote the image of a loss function $\ell$ by simply $\ell$. The figure shows how the superprediction set of a translation of the scaling of $\ell$ slides freely inside the $\mathrm{spr}(\ell_{\log})$. The bullet points are located at the image of $p \in \Delta^2$.

properties are determined by $\ell$ (see the relevant precise definitions below). We first discuss the case $n = 2$ (binary classification loss functions) since it is more intuitively understandable, and then the case $n \geq 2$. We summarize the main results as follows.

1. We recast the notion of a (strictly) proper loss as a geometric property of the loss itself rather than its superprediction set. That is, properness is no longer considered a parametrization dependent property, it is a statement about the geometric properties of the "loss surface" $\ell(\mathrm{relint}(\Delta^n))$ (the boundary of the superprediction set). See lemmas 2.7 and 3.2.

2. A geometric comparison is performed. For $n = 2$ in terms of the curvature of the "loss curves" (see Section 1.5 below), and for $n \geq 2$ in terms of the scalar second fundamental form of the "loss surfaces" (see Section 3 and Appendix A), which measure how they curve inside $\mathbb{R}^n$. The precise statements are given in Lemma 2.13 and Lemma 3.6. Intuitively, these results tell us how the superprediction set of $\ell$ sits inside the superprediction set of the log loss. We remark a fundamental difference between the cases $n = 2$ and $n \geq 2$: for $n = 2$, as Vovk (2015) observes, mixability can be related to the quotient of the curvature of $\ell$ and the curvature of $\ell_{\log}$. In this case the image of $\ell$ is a curve in $\mathbb{R}^2$ whose curvature can be easily computed. In higher dimensions the notion of curvature is far more complicated and different notions of curvature exist (see for example (Lee, 2018)). For $n \geq 2$, our geometric comparison relies on a comparison of the second fundamental forms of the "loss surfaces" defined by $\ell$ and $\ell_{\log}$ – the second fundamental form measures how a surfaces is curved inside an ambient space, which is $\mathbb{R}^n$ in our case (see Appendix A). In this regard, part of our work can be interpreted as a coordinate independent version of the main computations carried out in (van Erven et al., 2012).

3. Finally, we interpret our result from the point of view of convex analysis to give a new characterization of mixability. More precisely, we show that a (strictly) proper loss function $\ell$ is $\eta$-mixable if and only if the superprediction set of $\ell$ *slides freely* (see Definition 4.11) inside the superprediction set of the log loss.

As byproducts, we obtain a general way to define mixability with respect to a fixed (strictly) proper loss function, further properties and consequences for binary classification loss functions, particularly for composite losses and canonical links, and a bridge to the results obtained in (van Erven et al., 2012).

Since we treat loss functions from the point of view of differential geometry and convex geometry, a considerable background in these topics is needed. We present this work as self-contained as possible and spend some time providing the intuition and motivation for the results (and sometimes the background) which naturally results in a longer exposition. In Section 2 we treat the binary case, in Section 3 the multi-class case to obtain the geometric interpretation of properness and mixability and perform the geometric comparison (in terms of curvature). In Section 4 we make the connections to convex geometry and obtain the geometric characterization of mixability in terms of the sliding freely conditions of superprediction sets. In Section 5, we provide some additional comments and remarks, and provide a bridge between our results and those in van Erven et al. (2012).

### 1.3 Setup

Here we summarize our setup, for more details see van Erven et al. (2012). Denote by $[n]$ the set of natural numbers $\{1, ..., n\}$. The set of probability distributions on a finite set $\mathcal{Y}$ with $|\mathcal{Y}| = n \in \mathbb{N}$ is given by

$$\Delta^n = \left\{ (p_1, ..., p_n) \in \mathbb{R}^n \,\middle|\, \sum_{i=1}^n p_i = 1, p_i \geq 0, \text{ for } i = 1, ..., n \right\}. \tag{1.1}$$

We note that $\Delta^n$ is a manifold with (non-smooth) boundary of dimension $n-1$. Moreover, $\Delta^n$ is a hypersurface in $\mathbb{R}^n$; we denote the interior (as a manifold) of $\Delta^n$ as $\text{int}(\Delta^n)$ which is the same set as the *relative interior* $\text{relint}(\Delta^n)$ of $\Delta^n$. We define the *standard parametrization of* $\Delta^n$ as the map $\Phi_{\text{std}} \colon S_{n-1} \subset \mathbb{R}^{n-1} \longrightarrow \Delta^n$ given by

$$\Phi_{\text{std}}(t_1, ..., t_{n-1}) = \left( t_1, ..., t_{n-1}, 1 - \sum_{i=1}^{n-1} t_i \right), \tag{1.2}$$

where $S_{n-1} = \left\{ (t_1, ..., t_{n-1}) \in \mathbb{R}^{n-1} \,\middle|\, \sum_{i=1}^{n-1} t_i \leq 1, t_i \geq 0, \text{ for } i = 1, ..., n-1 \right\}$.

In particular, when $n = 2$ the standard parametrization of $\Delta^2$ is the map $\Phi_{\text{std}} \colon [0, 1] \longrightarrow \Delta^2$ given by $\Phi_{\text{std}}(t) = (t, 1 - t)$.

**Definition 1.2.** *A loss function is a map* $\ell \colon \Delta^n \times \mathcal{Y} \longrightarrow \mathbb{R}_{\geq 0}$ *such that for each* $k \in \mathcal{Y}$, *the map* $\ell(\cdot, k) \colon \Delta^n \longrightarrow \mathbb{R}$ *is continuous.*

Given a loss function $\ell$, $p \in \Delta^n$ and $k \in \mathcal{Y}$, the value $\ell(p, k)$ represents the penalty of predicting $p$ upon observing $k$. We define the *partial losses* of a loss function $\ell$ as the maps $\ell_i \colon \Delta^n \longrightarrow \mathbb{R}_{\geq 0}$ given by $\ell_i(p) = \ell(p, i)$. A loss function can be described in terms of its partial losses as

$$\ell(p, k) = \sum_{i=1}^n [\![ k = i ]\!] \ell_i(p).$$

Thus, we can identify a loss function $\ell$ with the map $\ell \colon \Delta^n \longrightarrow \mathbb{R}_{\geq 0}^n$ determined by its partial losses

$$\ell(p) = (\ell_1(p), ..., \ell_n(p)).$$

In this work we follow this convention unless stated otherwise. Note that this way we can see a loss function $\ell$ as an embedding of $\text{int}(\Delta^n)$ into $\mathbb{R}_{\geq 0}^n$ (assuming enough properties on $\ell$). We will see later that properness ensures the image of this embedding to be a nice hypersurface of $\mathbb{R}^n$ with appealing geometric properties. Under the assumption that the outcomes are distributed with probability $p \in \Delta^n$, we make the below definitions following (van Erven et al., 2012; Reid & Williamson, 2010).

**Definition 1.3.** *Given a loss function $\ell$, we define the* conditional risk *as the map $L : \Delta^n \times \Delta^n \longrightarrow \mathbb{R}$ as*

$$L(p, q) := \langle \ell(q), p \rangle,$$

*and the associated* conditional Bayes risk *as the map $\underline{L} : \Delta^n \longrightarrow \mathbb{R}$ given by*

$$\underline{L}(p) := \inf_{q \in \Delta^n} L(p, q) = \inf_{q \in \Delta^n} \langle \ell(q), p \rangle.$$

**Definition 1.4.** *A loss function $\ell \colon \Delta^n \longrightarrow \mathbb{R}^n_{\geq 0}$ is said to be* proper *if for any $p \in \Delta^n$*

$$p = \inf_{q \in \Delta^n} L(p, q)$$

*for all $q \in \Delta^n$. In other words, $L(p, \cdot)$ has a minimum at $p$. When $p$ is the only minimum of $L(p, \cdot)$ we say that $\ell$ is* strictly proper.

Proper losses are the natural losses to use when predicting class probabilities — they are such that perfect prediction is not penalized; for further discussion of why they are important, and additional references to the literature, see (Buja et al., 2005; Williamson & Cranko, 2022; Williamson et al., 2016).

For our geometric considerations it will be useful to denote the image of $\Delta^n$ under $\ell$ by $M_\ell$, and impose enough differentiability conditions on $\ell$ so that $M_\ell$ is (at least) a $C^2$-manifold. See Definitions 2.1 and 3.1 below.

We now recall the definition of *mixability* (see for example, (Vovk, 2015; van Erven et al., 2012)). For $\eta > 0$, let $E_\eta \colon \mathbb{R}^n \longrightarrow \mathbb{R}^n$ be the *$\eta$-exponential projection* defined as

$$E_\eta(y) := (e^{-\eta y_1}, ..., e^{-\eta y_n}). \tag{1.3}$$

A loss function $\ell$ is called *$\eta$-mixable* if the image of its *superprediction set*, $\mathrm{spr}(\ell)$, given by

$$\mathrm{spr}(\ell) := \{\lambda \in [0, \infty)^n \,|\, \text{there is } q \in \Delta^n \text{ such that } \ell_i(q) \leq \lambda_i \text{ for } i \in [n]\}, \tag{1.4}$$

is convex under the $\eta$-exponential projection, that is $E_\eta(\mathrm{spr}(\ell)) \subset [0, 1]^n$ is convex. We say that $\ell$ is *mixable* if $\ell$ is $\eta$-mixable for some $\eta > 0$.

**Definition 1.5.** *Let $\ell$ be a mixable loss function. The* mixability constant *of $\ell$, $\eta^*_\ell$, is defined as*

$$\eta^*_\ell := \sup_{\eta > 0} \{\eta > 0 \,|\, \ell \text{ is } \eta\text{-mixable}\}.$$

## 1.4 Motivation

In this part we mainly discuss the case $n = 2$ since it is more intuitively accessible. It has been made evident that there is a strong relation between properness and mixability. Here we make this relation more explicit and transparent from a geometric point of view. The basic motivation is as follows. It is commonly understood that properness is a property that depends on the parametrization of the boundary of the superprediction set of $\ell$ (Vovk, 2015). It has been also shown that it is related to the "curvature" of the Bayes risk, since it requires that the superprediction set remains convex under the $\eta$-exponential projection given by (1.3) (with the standard parametrization of the simplex $\Delta^2$) (Buja et al., 2005; Reid & Williamson, 2010; van Erven et al., 2012). Mixability is considered to be a stronger notion of convexity (Vovk, 2015), for some $\eta > 0$. The basic observation in this work is that it is possible recast properness from a geometric point of view, i.e., independent of the parametrization of $\Delta^n$. More precisely, we define properness in terms of the loss function viewed as a map $\ell \colon \Delta^n \longrightarrow \mathbb{R}^n_{\geq 0}$ rather than in terms of the superprediction set $\mathrm{spr}(\ell)$ (as it is usually defined). More precisely, to determine whether a given $\ell$ is proper or not, it is not enough to look at image $\ell(\Delta^n)$ (as the boundary of $\mathrm{spr}(\ell)$) but rather how $\Delta^n$ is mapped into $\mathbb{R}^n_{\geq 0}$ by $\ell$ — since we will be using tools of differential geometry, we will assume $C^2$ differentiability (see Section 2). More precisely, restricting first to $n = 2$ (see Lemma 2.7 below), a given loss function $\ell \colon \Delta^2 \longrightarrow \mathbb{R}^2_{\geq 0}$ will be (strictly) proper if and only if

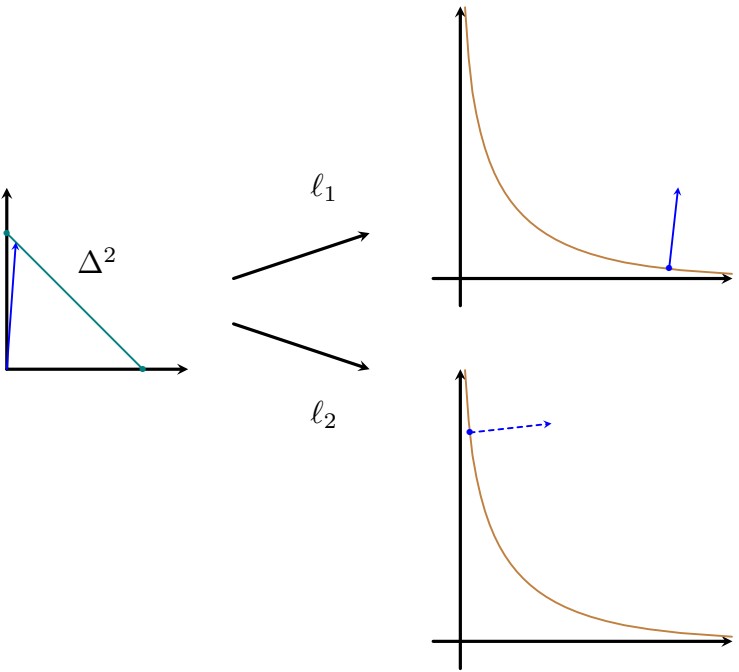

Figure 2: Consider the two loss functions given by $\ell_1(p_1, p_2) = (-\log(p_1), -\log(p_2))$ and $\ell_2(p_1, p_2) = (-\log(p_2), -\log(p_1))$, for $p = (p_1, p_2) \in \Delta^2$. Although $\mathrm{spr}(\ell_1) = \mathrm{spr}(\ell_2)$, $\ell_2$ is not proper since the normal vector at $\ell_2(p)$ is not $\pm p/|p|$ for any $p \in \Delta^2$.

1. the normal vector to $\ell(\Delta^n)$ at $\ell(p)$ is equal to $\pm p/|p|$ for all $p \in \mathrm{int}(\Delta^2)$, and

2. the curvature (see Section 1.5 below) at any point $\ell(p)$ with respect to the unit normal vector $\mathbf{n} = p/|p|$ is strictly positive for all $p \in \mathrm{int}(\Delta^2)$.

As observed in Figure 2, $\mathrm{spr}(\ell_1) = \mathrm{spr}(\ell_2)$, which implies that their boundaries coincide (as a set). In particular, this implies that it is possible to "parametrize" the boundary of $\ell_2(\Delta^2)$, $\partial(\ell_2(\Delta^2))$, in the same way as $\partial(\ell_1(\Delta^2))$ in order to have a proper loss. However, note that this changes the map $\ell_2$ and hence from the point of view of this work, this is a different loss function. In practice, one is given a loss function $\ell$ rather than a superprediction set $\mathrm{spr}(\ell)$, therefore we look at losses as individual maps from $\Delta^2$ to $\mathbb{R}^2_{\geq 0}$ instead of looking at their superpredictions sets and obtaining a proper loss by choosing a convenient parametrization of $\partial(\mathrm{spr}(\ell))$.

**Remark 1.6.** *Our strength by characterizing proper loss functions in this way is that we will be able to apply techniques from differential geometry, however, these considerations only work for loss functions which are sufficiently differentiable. For a general set up, it is possible to characterize properness of a loss function in a fairly simple way via the convexity of its superprediction set. More precisely, the "loss surface" is the subgradient of the support function of the superprediction set. This was thoroughly studied by Williamson & Cranko (2022). We briefly explore some connections to our work in Section 4. Alternative approaches to extending and better understanding mixability include (Reid et al., 2015) and (Mhammedi & Williamson, 2018).*

### 1.5   Comments about the curvature of planar curves

The second condition for $\ell$ to be proper mentioned above involves a condition on the curvature of $\ell(\mathrm{int}(\Delta^2))$. We now make this notion precise. Recall that if $\alpha(t) = (x_1(t), x_2(t))$ is a $C^2$ curve with $\alpha'(t) = (x_1'(t), x_2'(t)) \neq (0, 0)$ for all $t$ in its domain, then its curvature can be seen a measurement of the variation of its unit normal

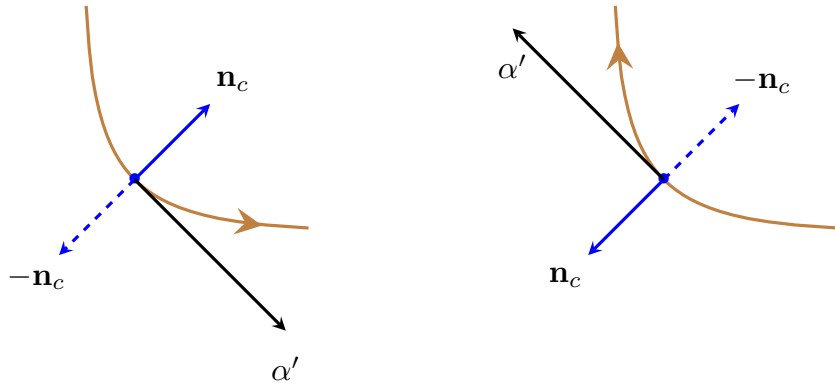

Figure 3: For a regular curve $\alpha$, at each point we have two normal unit vectors.

vector at each point. We define the *canonical normal vector at* $\alpha(t)$, $\mathbf{n}^c(t)$, as the unit normal vector in the direction obtained by rotating $\alpha'(t)$ 90° counterclockwise. Then, the *signed curvature of* $\alpha$ *at* $t$ is defined as

$$\kappa_\alpha(t) := \frac{x_1''(t)x_2'(t) - x_1'(t)x_2''(t)}{(x_1'(t)^2 + x_2'(t)^2)^{3/2}}. \tag{1.5}$$

The interpretation of this number is as follows: $\kappa_\alpha(t)$ is positive if $\alpha$ "curves" in the direction of $\mathbf{n}^c(t)$. However, note that at each point we have two normal vectors: $\pm\mathbf{n}^c(t)$. Thus, $\mathbf{n}^c(t)$ and $\kappa_\alpha$ depend on the direction of $\alpha$ (i.e., $\alpha'$), and their values differ by a negative sign. Thus, we can talk about the curvature of $\alpha$ with respect to a chosen unit vector $\mathbf{n}$ (either choosing $\mathbf{n}^c$ or $-\mathbf{n}^c$ for all points, assuming this is possible, which is the case for the curves we will consider here, see Figure 3) and denote it by $\kappa_\alpha^+$. In the case when $\mathbf{n} = \mathbf{n}^c$, then $\kappa_\alpha^+ = \kappa_\alpha$, and when $\mathbf{n} = -\mathbf{n}^c$, then $\kappa_\alpha^+ = -\kappa_\alpha$. Since $\kappa_\alpha$ is invariant under reparametrizations (up to a sign), we can simply talk at the curvature of $\alpha$ at a given point $p$ in the image of $\alpha$. In Section 2 we make precise our choice in (2) above. For a summary of geometry of curves see Appendix A.

Going back to loss functions, suppose $\ell : \Delta^2 \longrightarrow \mathbb{R}_{\geq 0}^2$ is a loss function. Since $\Delta^2$ is a 1-manifold, any parametrization around a point (of its interior) can be assumed to be of the form $\Phi : (a, b) \subset \mathbb{R} \longrightarrow \Delta^2$ for some $a < b$. Thus, the *local expression* of $\ell$ under this parametrization $\widetilde{\ell} = \ell \circ \Phi$ is a curve in $\mathbb{R}^2$. By changing $\Phi$ around the same point, we are *reparametrizing* $\widetilde{\ell}$. Since curvature is independent of coordinates (i.e., of the $\Phi$ used) up to a sign, we can define the curvature of the *loss curve* $\ell(\text{int}(\Delta^2))$ with respect to a chosen unit normal vector (which will depend only on $\ell$). To compute it from its definition in (1.5), we need to choose a parametrization $\Phi$, and as we will see, many times it is convenient to take $\Phi = \Phi_{\text{std}}$.

**Remark 1.7.** *One could avoid part of the technical complications above by choosing beforehand* $\Phi = \Phi_{\text{std}}$, *as it is usually implicitly done, and then requiring* $\ell_1$ *and* $\ell_2$ *to be monotone (cf. Buja et al. (2005); Reid & Williamson (2010); Shuford et al. (1966); Vovk (2015)) – essentially, this amounts to choosing "direction" for the admissible loss curves. Although this approach is appealing since the curve parameter (t in our case) can be directly interpreted as a probability, and moreover it simplifies calculations since in this case the convention can be chosen so that the signed curvature coincide with* $\kappa^+$ *(see for example Vovk (2015)), when considering the multi-class case, the notion of "direction" breaks down and it is not clear which properties of* $M_\ell = \ell(\Delta^n)$ *one should consider. The approach we consider here gives a concrete logical path to a generalization to the multi-class case (see Section 3).*

## 1.6 Reconciling this point of view with previous works

In this part we explain how to "translate" the results we obtain here to previous results regarding proper losses and mixability. We do this in particular with (Reid & Williamson, 2010) and (Vovk, 2015).

**(Reid & Williamson, 2010)** Let $\Phi = \Phi_{\mathrm{std}}$. The parameter $\widehat{\eta}$ in (Reid & Williamson, 2010) corresponds to the parameter $t$ here, $\ell_1(\widehat{\eta})$ and $\ell_{-1}(\widehat{\eta})$ correspond to $\widetilde{\ell}_1(t)$ and $\widetilde{\ell}_2(t)$, respectively. Although the regularity assumption in (Reid & Williamson, 2010) is initially only differentiability of the partial losses, when discussing the *weight* of a loss function they impose $C^2$ regularity. From (Reid & Williamson, 2010, Theorem 1), we see that a loss $\ell$ is proper if (in particular) $\ell'_{-1} > 0$ and $\ell'_0 < 0$. We can heuristically say that $\ell$ goes from "right" to "left". This means that in this case, $\kappa^+_\ell(\widehat{\eta}) = -\kappa_\ell(\widehat{\eta})$. The log loss in this case is $\ell_{\log}(\widehat{\eta}) = (-\ln(\widehat{\eta}), -\ln(1 - \widehat{\eta}))$.

**(Vovk, 2015)** In (Vovk, 2015) the loss functions are defined as maps $(\lambda_0(p), \lambda_1(p))$, with $\lambda_0$ increasing and $\lambda_1$ decreasing (infinite differentiable). In this case, heuristically, losses go from "left" to "right" so that $\kappa^+_\lambda(p) = \kappa_\lambda(p)$. To relate this convention to ours, we set $\Phi(t) = (1 - t, t)$. Then the parameter $p$ in (Vovk, 2015) corresponds to $t$ and $\lambda_0$ and $\lambda_1$ correspond to $\widetilde{\ell}_1$ and $\widetilde{\ell}_2$. The log loss is then given by $\lambda(p) = (-\ln(1 - p), -\ln(p))$.

Therefore, from our point of view, in previous works there is an implicit choice of a parametrization of $\Delta^2$, particularly motivated to interpret the parameter as a probability. However, sometimes this might not be the case and a *link function* is needed (Reid & Williamson, 2010) – this fits well with our approach as a link function for us is a different choice of parametrization; this will carefully explained in Section 2.7. In favor of the study of loss functions using tools from differential geometry we are then motivated to eliminate this choice of parametrization and consider $\ell$ as a map between manifolds (namely, $\mathrm{int}(\Delta^2)$ and $\ell(\mathrm{int}(\Delta^2))$ as a submanifold of $\mathbb{R}^2$). Although picking a general parametrization of $\Delta^2$ complicates the interpretation of the parameter, it makes other properties of loss functions transparent. This approach has, to the knowledge of the authors, never been explored. We remark that, however, one can always set $\Phi = \Phi_{\mathrm{std}}$ and reinterpret the results of this work as the parameter being a probability. With this geometric characterization of loss functions and properness at hand we continue to study mixability.

## 2 Properness and Mixability for Binary Classification

We first restrict our discussion to binary classification, i.e., setting $n = 2$. Thus, we consider maps $\ell \colon \Delta^2 \longrightarrow \mathbb{R}^2_{\geq 0}$, where $\Delta^2 = \{(p_1, p_2) \in \mathbb{R}^2 \mid p_1 + p_2 = 1\}$, with partial losses $\ell_1(p)$ and $\ell_2(p)$. In this case the standard parametrization of $\Delta^2$ is given by $\Phi_{\mathrm{std}}(t) = (t, 1 - t)$ for $t \in [0, 1]$. When a parametrization of $\Delta^2$, say $\Phi$, is chosen, then the local expression of $\ell$ with respect to $\Phi$ ($\widetilde{\ell} = \ell \circ \Phi$) is a map from some interval $I \subset \mathbb{R}$ to $\mathbb{R}^2$, that is, a curve in the plane $\mathbb{R}^2$.

Dating back to (Haussler et al., 1995; Vovk, 1998) it has been established that properness of a loss function imposes strong conditions on the first and second derivatives of their partial losses. In (Vovk, 2015) these relations were expressed by means of the curvature of the loss curve. Moreover, in (Buja et al., 2005; Reid & Williamson, 2010) properness is related to the second derivative of its Bayes risk, which in a way can be interpreted as its curvature. However, in these works there is always an implicit choice of parametrization of $\Delta^2$, which in turn imposes certain restrictions on the "admissible" loss functions, particularly making the results parametrization dependent. In this section, we first recast properness as a geometric property which allows us to obtain results in a parametrization (or coordinate) independent way.

**Definition 2.1.** *An* admissible loss function *is a map* $\ell \colon \Delta^2 \longrightarrow \mathbb{R}^2_{\geq 0}$ *such that*

(i) $\ell(\mathrm{int}(\Delta^2)) \subset \mathbb{R}^2_{\geq 0}$ *is a 1-manifold of class* $C^2$,

(ii) *there exists a* $C^1$ *map* $\mathbf{n} \colon \ell(\mathrm{int}(\Delta^2)) \to N\ell(\mathrm{int}(\Delta^2))$, $\mathbf{n}(\ell(p)) =: \mathbf{n}_{\ell(p)}$, *where* $N\ell(\mathrm{int}\Delta^2)$ *is the normal space of* $\ell(\mathrm{int}\Delta^2)$, *and*

(iii) $\mathbf{n}(\ell(p))$ *or* $-\mathbf{n}(\ell(p))$ *belongs to* $\mathbb{R}^2_{>0}$ *for all* $p \in \mathrm{int}(\Delta^2)$.

*We denote the set of admissible loss functions as* $\mathcal{L}$.

**Remark 2.2.** *We give the following interpretation of the previous definition. (i) simply says that the loss curve (once parametrized) is twice differentiable with continuous second partial derivatives. (ii) prevents some*

*"anomalies" on $\ell$, for example, $\ell$ can not be constant on a neighborhood of a point. (iii) defines a subfamily of loss curves which are not allowed to vary "too much". This definition should be compared to the definition of loss functions in (Vovk, 2015, Section 2).*

**Definition 2.3.** *Let $\ell \in \mathcal{L}$. Let $\mathbf{n} \colon \ell(\text{int}(\Delta^2)) \to N\ell(\text{int}(\Delta^2))$ be the map that assigns to each $\ell(p)$ the normal vector to $M_\ell$ at $\ell(p)$ that lies in $\mathbb{R}^2_{\geq 0}$. We denote by $\kappa^+_\alpha(\cdot)$ the signed curvature of $\alpha$ with respect to the unit normal belonging to $\mathbb{R}^2_{\geq 0}$. We refer to $\kappa^+_\alpha(\cdot)$ as the curvature with respect to the unit normal vector pointing towards $\mathbb{R}^2_{\geq 0}$.*

## 2.1 Proper losses

**Lemma 2.4.** *Suppose that $\ell$ in $\mathcal{L}$ is strictly proper, then the signed curvature of the loss curve $\ell(\Delta^2)$ has a sign. Moreover, its curvature, $\kappa_\ell$, is positive with respect to unit normal vector (field) pointing towards $\mathbb{R}^2_{\geq 0}$.*

*Proof.* Let $p_0 \in \text{int}(\Delta^2)$ and let $\Phi \colon I \subset \mathbb{R} \longrightarrow \Delta^2$ be a parametrization of $\Delta^2$ around $p_0 = \Phi(t_0)$, for some $t_0 \in I$, which we use to obtain a parametrization of $\Delta^2 \times \Delta^2$ around $(p_0, p_0)$[3]. We consider the local expression of $L$ given by

$$\widetilde{L}(t, s) = \langle \ell(\Phi(s)), \Phi(t) \rangle.$$

Using strict properness we know that fixing $t$, the function $\widetilde{L}(t, \cdot)$ achieves a minimum at $s = t$ (and it is the only one), that is

$$0 = \partial_s \widetilde{L}(t, s)|_{s=t} = \langle \widetilde{\ell}'(t), \Phi(t) \rangle = \widetilde{\ell}'_1(t)\Phi_1(t) + \widetilde{\ell}'_2(t)\Phi_2(t), \tag{2.1}$$

$$0 < \partial_{ss} \widetilde{L}(t, s)|_{s=t} = \langle \widetilde{\ell}''(t), \Phi(t) \rangle = \widetilde{\ell}''_1(t)\Phi_1(t) + \widetilde{\ell}''_2(t)\Phi_2(t). \tag{2.2}$$

To compute the sign of the signed curvature of $\ell(\Delta^2)$ it is enough to determine the sign of $\widetilde{\ell}'_1(t)\widetilde{\ell}''_2(t) - \widetilde{\ell}''_1(t)\widetilde{\ell}'_2(t)$. Without loss of generality, assuming $\Phi_2 \neq 0$ on this coordinate neighborhood we can write

$$\widetilde{\ell}'_1(t)\widetilde{\ell}''_2(t) - \widetilde{\ell}''_1(t)\widetilde{\ell}'_2(t) = \widetilde{\ell}'_1(t)\widetilde{\ell}''_2(t) - \widetilde{\ell}''_1(t)\left[-\frac{\widetilde{\ell}'_1(t)\Phi_1(t)}{\Phi_2(t)}\right]$$

$$= \frac{\widetilde{\ell}'_1(t)}{\Phi_2(t)}\left[\widetilde{\ell}''_2(t)\Phi_2(t) + \widetilde{\ell}''_1(t)\Phi_1(t)\right]$$

$$= \frac{\widetilde{\ell}'_1(t)}{\Phi_2(t)}\left[\langle \widetilde{\ell}''(t), \Phi(t) \rangle\right] > 0,$$

where we have used (2.1) and (2.2). Notice that if $\widetilde{\ell}'_1(t) = 0$ for some $t$ then necessarily $\widetilde{\ell}'_2(t) = 0$ by (2.1), which is impossible in $\mathcal{L}$. Therefore $\widetilde{\ell}'_1$ has a sign and this sign determines the sign of the signed curvature of $\ell(\Delta^2)$.

For the second statement, notice that again using (2.1) we know that $\widetilde{\ell}'_1$ and $\widetilde{\ell}'_2$ have different signs (and they do not change). If $\widetilde{\ell}'_1 > 0$, then that means that the first coordinate increases and the second decreases, hence $\mathbf{n}(t)$ points towards $\mathbb{R}^2_{\geq 0}$ and $\kappa_{\widetilde{\ell}} > 0$. If $\widetilde{\ell}'_1 < 0$, then we are in the opposite case and in this case $\mathbf{n}(t)$ points to $\mathbb{R}^2_{\geq 0}$ and $\kappa_{\widetilde{\ell}} < 0$, thus the signed curvature with respect to $-\mathbf{n}(t)$ (the unit normal pointing towards $\mathbb{R}^2_{\geq 0}$) is positive. □

From the proof of the previous theorem we obtain the following corollary.

**Corollary 2.5.** *Let $\ell \in \mathcal{L}$. If $\ell$ is proper, then $p \in \text{int}(\Delta^2)$ is normal to the loss curve $\ell(\Delta^2)$ at $\ell(p)$.*

*Proof.* It follows directly from (2.1), since for fixed $p \in \text{int}(\Delta^2)$, $\langle \ell(q), p \rangle$ attains a minimum at $p$. □

---

[3]Notice that this particular choice of coordinates around $(p_0, p_0)$ suffices since we want to conclude something about the curvature of the curve loss $\ell$.

**Lemma 2.6.** *In $\mathcal{L}$, proper implies strictly proper.*

*Proof.* Let $p \in \text{int}(\Delta^2)$, and suppose that there is $p^* \neq p \in \text{int}(\Delta^2)$, such that

$$\langle \ell(p^*), p \rangle = \inf_{q \in \Delta^2} \langle \ell(q), p \rangle.$$

Using (2.1), we see that $p^*$ is normal to $\ell$ at $\ell(p)$, and hence $p$ and $p^*$ are parallel. Since both belong to $\Delta^2$, it follows that $p^* = p$, which is a contradiction. $\qquad\square$

Therefore, in what follows (as long as we stay within $\mathcal{L}$) we will use proper and strictly proper interchangeably.

Note that the converse of Lemma 2.4 does not hold. That is, there are $\ell \in \mathcal{L}$ which have positive signed curvature (with respect to the unit normal pointing towards $\mathbb{R}^2_{\geq 0}$), but are not proper. Indeed let $\ell$ be defined as

$$\ell(p) = (-\ln(p_2), -\ln(p_1)).$$

Taking the (standard) parametrization $\Phi_{\text{std}}(t) = (t, 1 - t)$ we see that the loss curve $\widetilde{\ell}$ goes from left to right so $\mathbf{n}_{\widetilde{\ell}(t)}$ points towards $\mathbb{R}^2_{\geq 0}$. Moreover, we can readily see that the (signed) curvature $\kappa_{\widetilde{\ell}}$ is positive. However, $\Phi_{\text{std}}(t)$ is not normal to $\widetilde{\ell}$ at $\widetilde{\ell}(t)$, thus by Corollary 2.5, $\ell$ can not be proper.

Therefore, we obtain the following characterization of proper losses in $\mathcal{L}$.

**Lemma 2.7.** *Let $\ell \in \mathcal{L}$. Then $\ell$ is strictly proper if and only if $p$ is normal to the loss curve $\ell(\Delta^2)$ at $\ell(p)$ for all $p \in \text{int}(\Delta^2)$ and the signed curvature of $\ell(\Delta^2)$ with respect to the normal vector pointing towards $\mathbb{R}^2_{\geq 0}$ is positive at all points $\ell(p)$ for $p \in \text{int}(\Delta^2)$.*

*Proof.* The "if" part is Lemma 2.4. For the "only if" part, let $\ell \in \mathcal{L}$ be such that

$$\mathbf{n}_p = \pm \frac{p}{|p|}, \qquad (2.3)$$

$$\kappa^+_\ell > 0, \qquad (2.4)$$

where $\kappa^+_\ell$ is the signed curvature of $\ell$ with respect to the unit normal pointing towards $\mathbb{R}^2_{\geq 0}$. Let $p \in \text{int}(\Delta^2)$ and let $\Phi$ be a parametrization around $p$. We readily see that (2.3) implies that

$$\partial_s \widetilde{L}(t, s)|_{s=t} = 0,$$

while (2.4) implies $\partial_{ss} \widetilde{L}(t, s)|_{s=t} > 0$ by the proof of Lemma 2.4. This implies that fixing $t$, $\underline{\widetilde{L}}$ achieves its minimum at $s = t$. Then $\ell$ is proper and by Lemma 2.6, we conclude it is strictly proper. $\qquad\square$

**Remark 2.8.** *Notice that to check whether a given loss function $\ell \in \mathcal{L}$ is proper or not, it suffices to do it in any coordinate system. That is, given $\Phi$, we check conditions (2.3) and (2.4) for $\widetilde{\ell} = \ell \circ \Phi$.*

## 2.2 Mixable loss functions

We say that a loss function $\ell$ is *fair* if $\ell_1(p) \to 0$ as $p \to (0, 1)$ and $\ell_2(p) \to 0$ as $p \to (1, 0)$ (this is motivated by the interpretation when using the standard parametrization, see Reid & Williamson (2010)). In addition, recall that a loss function $\ell \in \mathcal{L}$ is proper if and only if

(i) $\mathbf{n}_{\ell(p)} = \frac{p}{|p|}$ can be chosen, and

(ii) $\kappa^+_\ell(p) > 0$

for all $p \in \text{int}(\Delta^2)$.

Thus, a prototype of a fair proper loss function is shown in Figure 4.

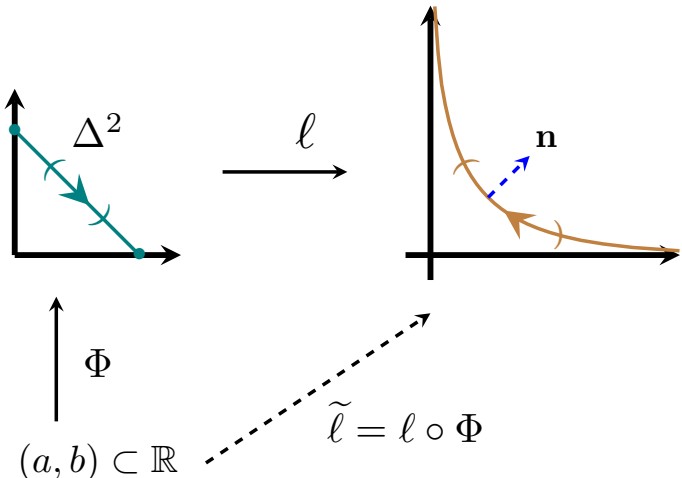

Figure 4: Prototype of a mixable fair proper loss function.

Recall from Section 1 that mixability is defined in terms of the superprediction set $\mathrm{spr}(\ell)$ of $\ell$. More precisely, for $\eta > 0$, consider the map

$$E_\eta(y_1, y_2) = \left(e^{-\eta y_1}, e^{-\eta y_2}\right),$$

where $E_\eta \colon \mathbb{R}^2_{\geq 0} \longrightarrow [0,1]^2$ is the exponential projection (1.3). Then, $\ell$ is $\eta$-mixable if and only if $E_\eta(\mathrm{spr}(\ell))$ is convex.

**Remark 2.9.** *We stress the fact that this definition depends on the superprediction set of $\ell$ rather than on $\ell$ itself – two different loss functions with the same superprediction set will be equally mixable. From our perspective, when talking about mixability of the map $\ell$ (i.e., without making reference to the superprediction set), we see that we can define it as follows. A loss $\ell$ is mixable if the 1-dimensional manifold $E_\eta \circ \ell(\mathrm{int}(\Delta^2))$ has signed curvature $\kappa^+_{E_\eta \circ \ell} \leq 0$. We will adopt the latter version here. Although clearly these definitions are equivalent, it is useful to have this at hand to relate mixability with properness. For now on, when we say $\ell$ is mixable we mean in the latter way. See Figure 5.*

We close this part by describing the log loss, which will play an important role. Let $\ell_{\log} \colon \Delta^2 \longrightarrow \mathbb{R}^2$, given by

$$\ell_{\log}(p) = \left(-\ln(p_1), -\ln(p_2)\right). \tag{2.5}$$

Let $\Phi = \Phi_{\mathrm{std}}$. Then

$$\widetilde{\ell}_{\log}(t) = \left(-\ln(t), -\ln(1-t)\right).$$

Since $\widetilde{\ell}'_{\log}(t) = \left(-t^{-1}, (1-t)^{-1}\right)$, its canonical normal vector is

$$\mathbf{n}_{\widetilde{\ell}_{\log}(t)} = -\frac{1}{\sqrt{t^2 + (1-t)^2}}\left((1-t)^{-1}, t^{-1}\right).$$

The curvature with respect to $-\mathbf{n}_{\widetilde{\ell}_{\log}(t)}$, the normal vector pointing towards $\mathbb{R}^2_{\geq 0}$, is then given by

$$\kappa^+_{\widetilde{\ell}_{\log}} = -\kappa_{\widetilde{\ell}_{\log}} = \frac{t(1-t)}{\left(t^2 + (1-t)^2\right)^{3/2}} > 0. \tag{2.6}$$

When there is no risk of confusion we denote $\kappa^+_{\widetilde{\ell}_{\log}}$ simply as $\kappa^+_{\log}$.

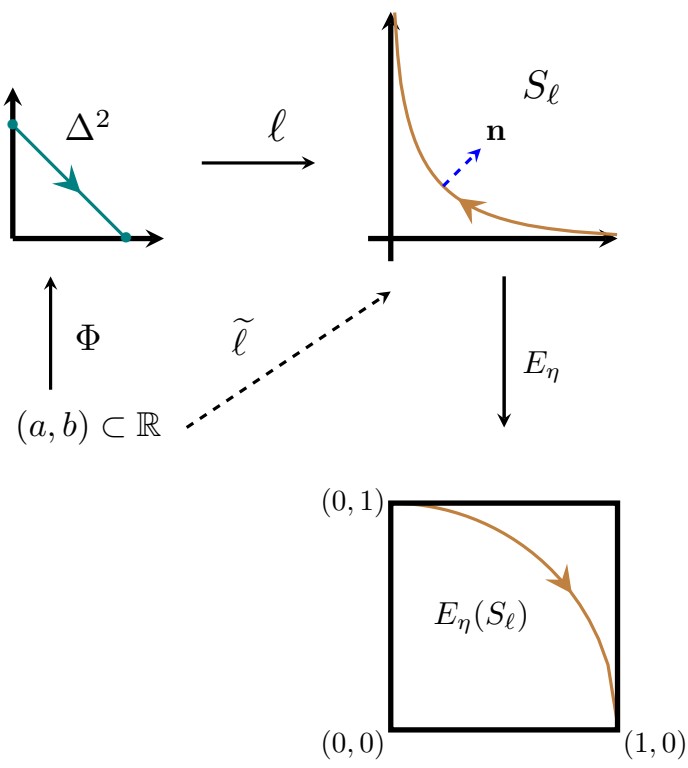

Figure 5: Diagram depicting how convexity of $E_\eta(\mathrm{spr}(\ell))$ is characterized by the principal curvatures of $E_\eta \circ \ell(\mathrm{int}(\Delta^2))$.

### 2.3 Mixability and curvature

Haussler et al. (1995) gave a characterization of the mixability constant of a mixable proper binary loss function $\ell$ in terms of the first and second derivatives of its partial losses. We reprove this characterization from a geometric point of view, that is, independent of the parametrization chosen for $\Delta^n$.

Let $\ell \in \mathcal{L}$ be proper and $\Phi$ a 1-chart parametrization[4] of $\Delta^2$, then $E_\eta(\mathrm{spr}(\ell))$ will be convex if and only if the curve $\gamma(t) = E(\ell(\Phi(t)))$ has negative curvature with respect to the unit normal pointing towards $\mathbb{R}^2_{\geq 0}$. Since $\ell$ is proper we can assume without loss of generality that $\kappa_\ell(p) = \kappa_\ell^+(p) > 0$. We are then interested in computing the signed curvature of

$$g(t) = (g_1(t), g_2(t)) = \left( E(\widetilde{\ell}_1(t)), E(\widetilde{\ell}_2(t)) \right) = \left( e^{-\eta \widetilde{\ell}_1(t)}, e^{-\eta \widetilde{\ell}_2(t)} \right),$$

and showing that $\kappa_g \geq 0$. We have

$$g_1'(t) = -\eta \widetilde{\ell}_1'(t) e^{-\eta \widetilde{\ell}_1(t)}$$
$$g_1''(t) = -\eta \widetilde{\ell}_1''(t) e^{-\eta \widetilde{\ell}_1(t)} + \eta^2 \widetilde{\ell}_1'(t)^2 e^{-\eta \widetilde{\ell}_1(t)}$$
$$= \eta e^{-\eta \widetilde{\ell}_1(t)} \left[ \eta \widetilde{\ell}_1'(t)^2 - \widetilde{\ell}_1''(t) \right]$$

and

$$g_2'(t) = -\eta \widetilde{\ell}_2'(t) e^{-\eta \widetilde{\ell}_2(t)}$$

---

[4]This means that the map $\Phi \colon D \longrightarrow \Delta^2$ is such that $\Phi(D) = \Delta^2$.

$$g_2''(t) = -\eta\widetilde{\ell}_2''(t)e^{-\eta\widetilde{\ell}_2(t)} + \eta^2\widetilde{\ell}_2'(t)^2 e^{-\eta\widetilde{\ell}_2(t)}$$
$$= \eta e^{-\eta\widetilde{\ell}_2(t)}\left[\eta\widetilde{\ell}_2'(t)^2 - \widetilde{\ell}_2''(t)\right],$$

and thus we have

$$\left(g_1'(t)^2 + g_2'(t)^2\right)^{3/2}\kappa_g(t) = -\eta\widetilde{\ell}_1'(t)e^{-\eta\widetilde{\ell}_1(t)}\eta e^{-\eta\widetilde{\ell}_2(t)}\left[\eta\widetilde{\ell}_2'(t)^2 - \widetilde{\ell}_2''(t)\right]$$
$$- \eta e^{-\eta\widetilde{\ell}_1(t)}\left[\eta\widetilde{\ell}_1'(t)^2 - \widetilde{\ell}_1''(t)\right]\left(-\eta\widetilde{\ell}_2'(t)e^{-\eta\widetilde{\ell}_0(t)}\right)$$
$$= \eta^2 e^{-\eta\widetilde{\ell}_1(t)}e^{-\eta\widetilde{\ell}_2(t)}\left[\widetilde{\ell}_1'(t)\widetilde{\ell}_2''(t) - \widetilde{\ell}_1'(t)\eta\widetilde{\ell}_2'(t)^2 + \widetilde{\ell}_2'(t)\eta\widetilde{\ell}_1'(t)^2 - \widetilde{\ell}_2'(t)\widetilde{\ell}_1''(t)\right]$$
$$= \eta^2 e^{-\eta\widetilde{\ell}_1(t)}e^{-\eta\widetilde{\ell}_2(t)}\left[\eta\widetilde{\ell}_2'(t)\widetilde{\ell}_1'(t)(\widetilde{\ell}_1'(t) - \widetilde{\ell}_2'(t)) + \left[\widetilde{\ell}_1'(t)\widetilde{\ell}_2''(t) - \widetilde{\ell}_2'(t)\widetilde{\ell}_1''(t)\right]\right].$$

Note that the sign of $\widetilde{\ell}_1'(t)\widetilde{\ell}_2''(t) - \widetilde{\ell}_2'(t)\widetilde{\ell}_1''(t)$ is the sign of $\kappa_{\widetilde{\ell}}$. If $\kappa_{\widetilde{\ell}}$ is positive, then one can check that $\widetilde{\ell}_1'(t) > 0$ and $\widetilde{\ell}_2'(t) < 0$, thus the first term in brackets is necessarily negative. Thus by making $\eta$ large $\kappa_g(t)$ will become negative. Then we want

$$\eta\widetilde{\ell}_2'(t)\widetilde{\ell}_1'(t)(\widetilde{\ell}_1'(t) - \widetilde{\ell}_2'(t)) + \left[\widetilde{\ell}_1'(t)\widetilde{\ell}_2''(t) - \widetilde{\ell}_2'(t)\widetilde{\ell}_1''(t)\right] \geq 0,$$

that is,

$$\eta \leq \frac{\widetilde{\ell}_1'(t)\widetilde{\ell}_2''(t) - \widetilde{\ell}_2'(t)\widetilde{\ell}_1''(t)}{(-\widetilde{\ell}_2'(t)\widetilde{\ell}_1'(t))(\widetilde{\ell}_1'(t) - \widetilde{\ell}_2'(t))}.$$

When considering the case when the signed curvature is negative, we have:

**Lemma 2.10.** *Suppose that $\ell \in \mathcal{L}$ is a proper loss function. Then, if $\ell$ is mixable, for any 1-chart parametrization $\Phi$ of $\Delta^2$, the mixability constant is given by*

$$\eta_\ell^* = \inf_{t \in \Phi^{-1}(\mathrm{int}(\Delta^2))}\left|\frac{\widetilde{\ell}_1'(t)\widetilde{\ell}_2''(t) - \widetilde{\ell}_2'(t)\widetilde{\ell}_1''(t)}{\widetilde{\ell}_1'(t)\widetilde{\ell}_2'(t)(\widetilde{\ell}_1'(t) - \widetilde{\ell}_2'(t))}\right|. \tag{2.7}$$

*Conversely, if (2.7) holds, then $\ell$ is mixable with mixability constant $\eta_\ell^*$.*

By the local nature of curvature, it would be possible to consider a "local version" of Lemma 2.10, which would characterize a "local" notion of mixability. This alternative will not be pursued here.

Vovk (2015) observed that mixability for proper losses is equivalent to a quotient of curvatures being bounded away from zero. For the reader's convenience we prove this statement. To recover Vovk's statement observe that the properties he imposes on the loss functions imply that $\kappa^+$ is the signed curvature (see Section 1.6).

**Lemma 2.11.** *A proper loss function $\ell \in \mathcal{L}$ is mixable if and only if*

$$\inf_p \frac{\kappa_\ell^+(p)}{\kappa_{\log}^+(p)} > 0,$$

*where $\kappa_{\log}^+$ denotes the curvature of $\ell_{\log}$. Moreover, when this holds,*

$$\eta_\ell^* = \inf_p \frac{\kappa_\ell^+(p)}{\kappa_{\log}^+(p)} > 0,$$

*Proof.* By Lemma 2.10, $\ell$ is proper with mixability constant $\eta_\ell^* > 0$ if and only if

$$\eta_\ell^* = \inf_{t \in \Phi^{-1}(\mathrm{int}(\Delta^2))}\left|\frac{\widetilde{\ell}_1'(t)\widetilde{\ell}_2''(t) - \widetilde{\ell}_2'(t)\widetilde{\ell}_1''(t)}{\widetilde{\ell}_2'(t)\widetilde{\ell}_1'(t)(\widetilde{\ell}_1'(t) - \widetilde{\ell}_2'(t))}\right|,$$

for any given 1-chart parametrization $\Phi$. Setting $\Phi = \Phi_{\text{std}}$ and using (2.6), we have the following. For any $t \in \Phi^{-1}(\text{int}(\Delta^2))$,

$$
\left| \frac{\widetilde{\ell}_1'(t)\widetilde{\ell}_2''(t) - \widetilde{\ell}_2'(t)\widetilde{\ell}_1''(t)}{\widetilde{\ell}_2'(t)\widetilde{\ell}_1'(t)(\widetilde{\ell}_1'(t) - \widetilde{\ell}_2'(t))} \right|
$$

$$
= \left| \frac{\widetilde{\ell}_1'(t)\widetilde{\ell}_2''(t) - \widetilde{\ell}_2'(t)\widetilde{\ell}_1''(t)}{\left(\widetilde{\ell}_1'(t)^2 + \widetilde{\ell}_2'(t)^2\right)^{3/2}} \right| \frac{\left(\widetilde{\ell}_1'(t)^2 + \widetilde{\ell}_2'(t)^2\right)^{3/2}}{\left(\frac{1}{(1-t)^2} + \frac{1}{t^2}\right)^{3/2}} \frac{\left(\frac{1}{(1-t)^2} + \frac{1}{t^2}\right)^{3/2}}{\frac{1}{(1-t)^2}\frac{1}{t^2}} \frac{\frac{1}{(1-t)^2}\frac{1}{t^2}}{|\widetilde{\ell}_2'(t)\widetilde{\ell}_1'(t)(\widetilde{\ell}_1'(t) - \widetilde{\ell}_2'(t))|}
$$

$$
= \frac{\kappa_{\widetilde{\ell}}^+(t)}{\kappa_{\widetilde{\ell}_{\log}}^+(t)} \left( \frac{\widetilde{\ell}_1'(t)^2 + \widetilde{\ell}_2'(t)^2}{\frac{1}{(1-t)^2} + \frac{1}{t^2}} \right)^{3/2} \frac{\frac{1}{(1-t)^2}\frac{1}{t^2}}{|\widetilde{\ell}_2'(t)\widetilde{\ell}_1'(t)(\widetilde{\ell}_1'(t) - \widetilde{\ell}_2'(t))|}
$$

$$
= \frac{\kappa_{\widetilde{\ell}}^+(t)}{\kappa_{\widetilde{\ell}_{\log}}^+(t)} \left( \frac{\widetilde{\ell}_1'(t)^2 \left(1 + \frac{t^2}{(1-t)^2}\right)}{\frac{t^2 + (1-t)^2}{t^2(1-t)^2}} \right)^{3/2} \frac{\frac{1}{(1-t)^2}\frac{1}{t^2}}{|\frac{t}{1-t}\widetilde{\ell}_1'(t)^3(1 + \frac{t}{1-t})|}
$$

$$
= \frac{\kappa_{\widetilde{\ell}}^+(t)}{\kappa_{\widetilde{\ell}_{\log}}^+(t)},
$$

where we used that by properness $\widetilde{\ell}_2'(t) = -\frac{t}{1-t}\widetilde{\ell}_1'(t)$ (see (2.1)).

Since $\kappa^+$ is independent of the parametrization, we obtain the result. $\qquad\square$

**Remark 2.12.** *Lemma 2.11 exemplifies the usefulness of $\Phi_{\text{std}}$. The curvature of $\ell_{\log}$ is easily computed with respect to the standard parametrization, by fixing $\Phi = \Phi_{\text{std}}$ we can easily recognize when the curvature of $\ell_{\log}$ appears in our computation. However, since curvature is a geometric quantity we know this relation between curvatures will hold for any other parametrization as well.*

Using this point of view, the following observations enlighten why the weight function in (Buja et al., 2005) and in (Reid & Williamson, 2010) basically encodes all the relevant information in the binary case. Recall that given a proper loss function $\ell$, the weight of $\ell$ (with respect to a local parametrization $\Phi$ of $\Delta^2$) is defined as

$$
w_{\ell_\Phi}(t) = \left| \frac{\widetilde{\ell}_1'(t)}{\Phi_2(t)} \right| = \left| \frac{\widetilde{\ell}_2'(t)}{\Phi_1(t)} \right|. \tag{2.8}
$$

We stress that the weight depends on the coordinates $\Phi$ of $\Delta$ that we use, and hence we use the notation $\ell_\Phi$. As observed in Remark 2.12, we sometimes set $\Phi = \Phi_{\text{std}}$ (as it is done in (Buja et al., 2005; Reid & Williamson, 2010)) to be able to recognize some terms.

**Lemma 2.13.** *Let $\ell \in \mathcal{L}$ be a proper loss and $\Phi$ a local parametrization of $\Delta^2$, denote by $\widetilde{\ell}_\Phi$ its local expression and by $w_{\ell_\Phi}$ be its weight. Then we have for any $t \in \Phi^{-1}(\text{int}(\Delta^2))$,*

$$
k_{\widetilde{\ell}_\Phi}^+(t) = \frac{1}{w_{\ell_\Phi}(t)}|\Phi_1'(t)| \left( \frac{1}{\Phi_1(t)^2 + \Phi_2(t)^2} \right)^{3/2}
$$

*and moreover, if $\lambda$ is another proper loss,*

$$
\frac{\kappa_{\widetilde{\ell}_\Phi}^+(t)}{\kappa_{\widetilde{\lambda}_\Phi}^+(t)} = \frac{w_{\lambda_\Phi}(t)}{w_{\widetilde{\ell}_\Phi}(t)}. \tag{2.9}
$$

*In particular, when $\Phi = \Phi_{\text{std}}$,*

$$
k_{\widetilde{\ell}_{\text{std}}}^+(t) = \frac{1}{w_{\ell_{\text{std}}}(t)} \left( \frac{1}{t^2 + (1-t)^2} \right)^{3/2}.
$$

*and if in addition, $\lambda = \ell_{\log}$ (with $\Phi = \Phi_{\text{std}}$),*

$$\frac{k^+_{\widetilde{\ell}_{\text{std}}}(t)}{\kappa^+_{\widetilde{\ell}_{\log}}(t)} = \frac{1}{w_{\ell_{\text{std}}}(t)} \frac{1}{t(1-t)} = \frac{w_{\ell_{\log}}(t)}{w_{\ell_{\text{std}}}(t)}. \tag{2.10}$$

*Proof.* Let $\ell \colon \Delta^2 \longrightarrow \mathbb{R}^2_{\geq 2}$ be a proper loss and let $\Phi$ be any parametrization of $\Delta^2$ around $p$. Let us compute $\kappa^+_{\widetilde{\ell}_\Phi}$ (assuming w.l.o.g. that $\kappa^+_{\widetilde{\ell}_\Phi} = \kappa_{\widetilde{\ell}_\Phi}$, which means $\widetilde{\ell}'_1 > 0$ and $\Phi'_1 < 0$).

$$\begin{aligned}
\kappa^+_{\widetilde{\ell}_\Phi}(t) &= \frac{\widetilde{\ell}'_1(t)\widetilde{\ell}''_2(t) - \widetilde{\ell}''_1(t)\widetilde{\ell}'_2(t)}{\left(\widetilde{\ell}'_2(t)^2 + \widetilde{\ell}'_1(t)^2\right)^{3/2}}\\
&= \left(\widetilde{\ell}'_1(t)\widetilde{\ell}''_2(t) + \widetilde{\ell}''_1(t)\widetilde{\ell}'_1(t)\frac{\Phi_1(t)}{\Phi_2(t)}\right)\left(\frac{\Phi_1(t)^2}{\Phi_2(t)^2}\widetilde{\ell}'_1(t)^2 + \widetilde{\ell}'_1(t)^2\right)^{-3/2}\\
&= \frac{\widetilde{\ell}'_1(t)}{\Phi_2(t)}\frac{1}{\widetilde{\ell}'_1(t)^3}\left(\Phi_1(t)\widetilde{\ell}''_1(t) + \Phi_2(t)\widetilde{\ell}''_2(t)\right)\left(\frac{\Phi_1(t)^2}{\Phi_2(t)^2} + 1\right)^{-3/2}\\
&= -\frac{\widetilde{\ell}'_1(t)}{\Phi_2(t)}\frac{1}{\widetilde{\ell}'_1(t)^3}\left(\Phi'_1(t)\widetilde{\ell}'_1(t) + \Phi'_2(t)\widetilde{\ell}'_2(t)\right)\left(\frac{\Phi_1(t)^2}{\Phi_2(t)^2} + 1\right)^{-3/2}\\
&= -\frac{\widetilde{\ell}'_1(t)}{\Phi_2(t)}\frac{1}{\widetilde{\ell}'_1(t)^3}\left(\Phi'_1(t)\widetilde{\ell}'_1(t) - \Phi'_2(t)\frac{\Phi_1(t)}{\Phi_2(t)}\widetilde{\ell}'_1(t)\right)\left(\frac{\Phi_1(t)^2}{\Phi_2(t)^2} + 1\right)^{-3/2}\\
&= -\frac{1}{\Phi_2(t)}\frac{1}{\widetilde{\ell}'_1(t)}\left(\Phi'_1(t) - \Phi'_2(t)\frac{\Phi_1(t)}{\Phi_2(t)}\right)\left(\frac{\Phi_1(t)^2 + \Phi_2(t)^2}{\Phi_2(t)^2}\right)^{-3/2}\\
&= -\frac{1}{\Phi_2(t)}\frac{1}{\widetilde{\ell}'_1(t)}\left(\frac{\Phi'_1(t)\Phi_2(t) - \Phi'_2(t)\Phi_1(t)}{\Phi_2(t)}\right)\left(\frac{\Phi_2(t)^2}{\Phi_1(t)^2 + \Phi_2(t)^2}\right)^{3/2}\\
&= -\frac{\Phi_2(t)}{\widetilde{\ell}'_1(t)}\left(\Phi'_1(t)\Phi_2(t) - \Phi'_2(t)\Phi_1(t)\right)\left(\frac{1}{\Phi_1(t)^2 + \Phi_2(t)^2}\right)^{3/2}\\
&= -\frac{1}{w_{\ell_\Phi}(t)}\left(\Phi'_1(t)\Phi_2(t) + \Phi'_1(t)\Phi_1(t)\right)\left(\frac{1}{\Phi_1(t)^2 + \Phi_2(t)^2}\right)^{3/2}\\
&= \frac{1}{w_{\ell_\Phi}(t)}\left(-\Phi'_1(t)\right)\left(\Phi_2(t) + \Phi_1(t)\right)\left(\frac{1}{\Phi_1(t)^2 + \Phi_2(t)^2}\right)^{3/2},
\end{aligned}$$

where we have used that by properness we know that $\langle\widetilde{\ell}'(t), \Phi(t)\rangle = 0$ ((2.1)), which implies $\langle\widetilde{\ell}''(t), \Phi(t)\rangle = -\langle\widetilde{\ell}'(t), \Phi'(t)\rangle$ by differentiating with respect to $t$ from the third to the fourth equality, and that $\Phi'_1(t) + \Phi'_2(t) = 0$ since $\Phi(t) \in \Delta^2$ from the third to last to the second to last equality.

Notice that in the last equation of the previous string of equalities, the only term involving $\ell$ is $\widetilde{\ell}'_1$ (or more precisely $w_{\ell_\Phi}(t)$) and the remaining terms depend only on the parametrization $\Phi$. Then we obtain

$$\frac{\kappa^+_{\widetilde{\ell}_\Phi}(t)}{\kappa^+_{\widetilde{\lambda}_\Phi}(t)} = \frac{w_{\lambda_\Phi}(t)}{w_{\ell_\Phi}(t)}.$$

The remaining statements follow from setting $\Phi = \Phi_{\text{std}}$ and (2.6). $\qquad\square$

**Remark 2.14.** *Combining Lemma 2.11 and (2.10), we recover the characterization of the mixability constant in terms of the quotient of weights obtained by van Erven et al. (2012, Section 4.1). However for the corresponding statement involving the quotient of second derivatives of the Bayes risks, the fact that $\Delta^2$ has*

*an affine parametrization is important. Indeed, this relies on Corollary 3 in (Reid & Williamson, 2010) that states that $w(t) = -\widetilde{\underline{L}}''(t)$. In general, it can be checked that*

$$\widetilde{\underline{L}}''(t) - \left[\frac{\Phi_1''(t)}{\Phi_1(t)}\right]\widetilde{\underline{L}}'(t) = -\Phi_2(t)\widetilde{\ell}_1'(t)^2\left(1 + \frac{\Phi_2(t)^2}{\Phi_1(t)^2}\right)^{3/2}\kappa_{\widetilde{\ell}}(t),$$

*which reduces to $w(t) = -\widetilde{\underline{L}}''(t)$ when $\Phi = \Phi_{\mathrm{std}}$. From the point of view of the present work, $\underline{L}$ (or a quotient of them) is not a good quantity to consider since it strongly depends on coordinates. However, notice that if one restricts to affine parametrizations of $\Delta^2$ then $\widetilde{\underline{L}}''(t)$ depends on $\widetilde{\ell}_1'(t)^2$ and $\kappa_{\widetilde{\ell}}(t)$ and hence in view of Lemma 2.13 restricting to a fixed affine parametrization of $\Delta^2$ will make quotients of the second derivative of the Bayes risk well behaved.*

Let us remark some points about Lemma 2.13.

- Let $\ell\colon \Delta^2 \longrightarrow \mathbb{R}$ be a given strictly proper, fair, loss function. Given a parametrization, we obtain a weight $w_{\ell_\Phi}$ given by (2.8), that is, the weight depends on the parametrization.
- The curvature of $\ell$ is independent of $\Phi$ up to a sign. However, when defining $\kappa_\ell^+$ we made the choice of the sign in a uniform way, thus the curvature is independent of the parametrization for the family of losses considered here. Then it follows that the quotient of curvatures is independent of the parametrization and by (2.9), it also follows that the quotient of weights is also independent of the coordinates (despite the weights being coordinate dependent themselves).
- A corresponding notion of weight in higher dimensions (for the multi-class case) is way more complicated and it is unclear whether using them would lead to successful results. One higher dimensional analog of curvatures is readily seen to be the so called "principal curvatures" of a hypersurface in Euclidean space (see Appendix A). This will be the main motivation when dealing with the multi-class case (Section 3) Alternative ways to characterize proper higher dimensional loss functions have been studied in (Williamson et al., 2016).

## 2.4 Geometric comparison of loss functions

Fix a proper, fair loss function $\lambda\colon \Delta^2 \longrightarrow \mathbb{R}^2$. Given another proper, fair loss function $\ell \neq \lambda$, how might we compare them? From the point of view of differential geometry, since given $p$ the normal vectors at $\lambda(p)$ and $\ell(p)$ coincide, it is natural to look at their curvatures. Motivated by Lemma 2.11, we impose (for the moment) the condition

$$\inf_{p\in\Delta^2}\frac{\kappa_\ell^+(p)}{\kappa_\lambda^+(p)} = 1.$$

Note that this implies that $\kappa_\ell^+(p) \geq \kappa_\lambda^+(p)$ for all $p \in \Delta^2$. We divide the comparison in steps for clarity.

(1) **Expressing $\lambda(\Delta^2)$ as a function.** Note that since $\lambda$ is proper and fair, the normal vector to a point $\lambda(p)$ can only be $(1, 0)$ when $p = (1, 0)$ (i.e., when evaluating $\lambda$ at the boundary of $\Delta^2$). Thus, the set $\lambda(\mathrm{int}(\Delta^2))$ can be expressed as a graph over the $x$-axis. To obtain an explicit expression let $\Phi = \Phi_{\mathrm{std}}$. We use the fact that $\widetilde{\lambda}_1\colon (0,1) \longrightarrow (0, l_1)$ (where $l_1$ could be infinity) is invertible. Then, we have that

$$\lambda(\mathrm{int}(\Delta^2)) = \{(x, f(x))\,|\,x \in (0, l_1)\}$$

where $f(x) = \lambda_2(\widetilde{\lambda}_1^{-1}(x), 1 - \widetilde{\lambda}_1^{-1}(x))$.

(2) **Translating and parametrizing $\ell(\Delta^2)$.** Let $p_0 \in \mathrm{int}(\Delta^2)$ with $\kappa_\ell^+(p_0) > \kappa_\lambda^+(p_0)$, if such $p_0$ does not exist then $\ell = \lambda$. We define $\ell^0\colon \Delta^2 \longrightarrow \mathbb{R}^2$ by $\ell^0(p) = \ell(p) + [\lambda(p_0) - \ell(p_0)]$, i.e., we translate $\ell$ so that it coincides with $\lambda$ at $\lambda(p_0)$. ($\ell_0$ is not fair anymore, however, the curvature is invariant under translations.)

We now parametrize $\ell(\Delta^2)$ as the graph of a function $g$ defined on an interval $I_0$ around $x_0$ (the $x$-coordinate of $\lambda(p_0)$), "aligning" it with $\lambda$ (we can assume this interval to be maximal). We let $g(x) = \ell_2^0((\widetilde{\ell}_1^0)^{-1}(x), 1 - (\widetilde{\ell}_1^0)^{-1}(x))$. Since $\kappa_\ell^+(p_0) > \kappa_\lambda^+(p_0)$, we know that around $x_0$ the graph of $g$ is to the northeast of $f$.

(3) **Comparison.** If the graph of $g$ is to the northeast of $f$ on the whole $I_0$, then we see that the superprediction set of $\ell^0$ is contained in that of $\lambda$. If this does not hold, it means that there is $x_1 \in I_0$ such that $f(x_1) = g(x_1)$, and w.l.o.g. we can assume $x_1 > x_0$. Thus we know that $g(x) - f(x) \geq 0$ on $[x_0, x_1]$ and $g(x) - f(x) = 0$ on $\{x_0, x_1\}$, i.e., the boundary of $[x_0, x_1]$. Define the second order operator which computes the curvature of the graph $(x, h(x))$ (see (A.1)):

$$L(h)(x) = \kappa_h^+(x) = \frac{1}{(1 + h'(x)^2)^{3/2}} h''(x).$$

Since $\kappa_\ell^+(p_0) > \kappa_\lambda^+(p_0)$, we see that $L(g - f) \geq 0$ on $[x_0, x_1]$. The maximum principle now implies that the supremum of $g - f$ is attained at the boundary on $[x_0, x_1]$, and hence we know that $f(x) = g(x)$ on $[x_0, x_1]$, which is a contradiction. Thus the superprediction set $\ell$ is contained in the superprediction set of $\lambda$ (see Section 4).

More generally, if we assume instead that

$$\inf_{p \in \Delta^2} \frac{\kappa_\ell^+(p)}{\kappa_\lambda^+(p)} = \eta,$$

for some $\eta > 0$, we see that (see Appendix A) that $\ell_\eta(p) = \eta\ell(p)$ satisfies

$$\inf_{p \in \Delta^2} \frac{\kappa_{\ell_\eta}^+(p)}{\kappa_\lambda^+(p)} = 1.$$

That is, we can reproduce the previous analysis with $\ell_\eta$ instead of $\ell$.

The previous discussion motivates right away a comparison between proper, fair loss functions.

**Definition 2.15.** *Let $\lambda \colon \Delta^2 \longrightarrow \mathbb{R}_{\geq 0}^2$ be a proper, fair loss in $\mathcal{L}$, which we call a base loss. We say that a proper, fair loss $\ell \colon \Delta^2 \longrightarrow \mathbb{R}^2$ is mixable with respect to $\lambda$ if*

$$\inf_{p \in \Delta^2} \frac{\kappa_\ell^+(p)}{\kappa_\lambda^+(p)} > 0.$$

## 2.5 Mixability and fundamentality as comparison to the log loss

Now, suppose $\ell \in \mathcal{L}$ is proper and fair. Thus, in particular $\kappa_\ell^+(p) > 0$ for all $p \in \text{int}(\Delta^2)$. We want to think of mixability as a geometric comparison to the log loss as suggested by Vovk (2015) and give a detailed interpretation of this comparison. We fix the standard parametrization of $\Delta^2$, $\Phi = \Phi_{\text{std}} \colon [0, 1] \longrightarrow \Delta^2$, given by

$$\Phi(t) = (t, 1 - t).$$

The log loss in these coordinates is thus given by

$$\widetilde{\ell}_{\log}(t) = (-\ln(t), -\ln(1 - t)),$$

and by (2.6), its curvature with respect to the unit normal pointing towards $\mathbb{R}_{\geq 0}^2$ is given by

$$\kappa_{\widetilde{\ell}_{\log}}^+(t) = \frac{t(1 - t)}{(t^2 + (1 - t)^2)^{3/2}}.$$

Notice that $\kappa^{+}_{\widetilde{\ell}_{\log}}(t) > 0$ for all $t \in (0,1)$ and $\kappa^{+}_{\widetilde{\ell}_{\log}}(t) \to 0$ as $t \to 0$ or $t \to 1$. Thus, clearly by Lemma 2.7, for any proper subinterval $C$ of $[0,1]$ (cf. (Vovk, 2015, Corollary 2)), we have

$$\inf_{t \in C} \frac{\kappa^{+}_{\ell}(t)}{\kappa^{+}_{\widetilde{\ell}_{\log}}(t)} > 0.$$

Thus, whether a proper, fair loss function $\ell$ is mixable or not will depend of the behavior of the quotient $\kappa^{+}_{\ell}(p)/\kappa^{+}_{\log}(p)$ as $p$ approaches $(0,1)$ and $(1,0)$. More precisely, we have obtained the following.

**Lemma 2.16.** *Let $\ell \in \mathcal{L}$ be a proper loss. Then $\ell$ is mixable if and only if*

$$\lim_{p \to (0,1)} \frac{\kappa^{+}_{\ell}(p)}{\kappa^{+}_{\log}(p)} > 0, \text{ and}$$

$$\lim_{p \to (1,0)} \frac{\kappa^{+}_{\ell}(p)}{\kappa^{+}_{\log}(p)} > 0.$$

Motivated by this we make the following definition.

**Definition 2.17.** *Let $\ell$ be a proper, fair loss function in $\mathcal{L}$, and $\Phi = \Phi_{\mathrm{std}}$ be the standard parametrization of $\Delta^2$. We say that is $\ell$ $(B_1, B_2)$-logarithmic at the boundary if*

$$\lim_{t \to 0^+} \frac{\kappa^{+}_{\widetilde{\ell}}(t)}{\kappa^{+}_{\widetilde{\ell}_{\log}}(t)} = B_1^{-1} > 0, \text{ and}$$

$$\lim_{t \to 1^-} \frac{\kappa^{+}_{\widetilde{\ell}}(t)}{\kappa^{+}_{\widetilde{\ell}_{\log}}(t)} = B_2^{-1} > 0.$$

Let us analyze what this means. Suppose that $\ell$ is proper and $(B_1, B_2)$-logarithmic. Then for any $t \in (0,1)$, using (2.10) in Lemma 2.13 and (2.8), we have

$$\frac{\kappa^{+}_{\widetilde{\ell}}(t)}{\kappa^{+}_{\widetilde{\ell}_{\log}}(t)} = \frac{1}{w_{\widetilde{\ell}_{\mathrm{std}}}(t)} \frac{1}{t(1-t)} = \left| \frac{1}{\widetilde{\ell}'_1(t)} \right| \frac{1}{t}.$$

Notice that as $t \to 0^+$,

$$B_1^{-1} = \lim_{t \to 0^+} \frac{1}{t} \frac{1}{|\widetilde{\ell}'_1(t)|} = \lim_{t \to 0^+} \left| \frac{(\ell_{\log})'_1(t)}{\widetilde{\ell}'_1(t)} \right|.$$

and similarly,

$$B_2^{-1} = \lim_{t \to 1^-} \frac{1}{1-t} \frac{1}{|\widetilde{\ell}'_2(t)|} = \lim_{t \to 1^-} \left| \frac{(\ell_{\log})'_2(t)}{\widetilde{\ell}'_2(t)} \right|.$$

that is, we are only comparing the rate at which $\ell_i$, $i = 1, 2$, go to 0 (since they do by fairness) with the rate at which the log loss does.

In (Vovk, 2015), Vovk defines a loss function $\lambda^*$ to be *fundamental* if given a (computable, proper, mixable) loss function $\lambda$ and a data sequence in $\zeta \in \mathbb{Z}^\infty$ that is random under $\lambda^*$ with respect to a prediction algorithm $F$, then it is random under $\lambda$ with respect to $F$. He shows that a fair, mixable $\ell \in \mathcal{L}$ is fundamental if and only if (using the notation in (Vovk, 2015))

$$\sup_{p \in [0,1]} \frac{\kappa_{\ell}(p)}{\kappa_{\log}(p)} < \infty.$$

Since we have seen that mixability can be regarded as a comparison of curvatures of the loss curve of $\ell$ and that of $\ell_{\log}$ and we have reinterpreted fundamentabiliy as a comparison of $\ell$ and $\ell_{\log}$ near the boundary building on Definition 2.15, we can easily come up with a notion of $\lambda$-fundamentality.

**Definition 2.18.** *Let $\lambda$ be a proper, fair loss function in $\mathcal{L}$. We say that a proper, fair loss function $\ell \in \mathcal{L}$ is $\lambda$-fundamental if*

- *$\ell$ is mixable with respect to $\lambda$, and*
- *when $\Phi = \Phi_{\mathrm{std}}$, we have*

$$\lim_{t \to 0^+} \frac{\kappa_{\widetilde{\ell}}^+(t)}{\kappa_{\widetilde{\lambda}}^+(t)} < \infty$$

$$\lim_{t \to 1^-} \frac{\kappa_{\widetilde{\ell}}^+(t)}{\kappa_{\widetilde{\lambda}}^+(t)} < \infty.$$

Suppose now that a mixable loss function $\ell \in \mathcal{L}$ is fundamental. Then there exist $\eta, \gamma > 0$ such that

$$\eta \le \frac{\kappa_\ell^+(p)}{\kappa_{\log}^+(p)} \le \gamma,$$

for all $p \in \mathrm{int}(\Delta^2)$. This implies that

$$\eta^{-1}\kappa_\ell^+(p) \ge \kappa_{\log}^+(p) \quad \text{and} \quad \kappa_{\log}^+(p) \ge \gamma^{-1}\kappa_\ell^+(p),$$

for all $p \in \mathrm{int}(\Delta^2)$, which readily implies (Appendix A) that

$$\kappa_{\eta\ell}^+(p) \ge \kappa_{\log}^+(p) \quad \text{and} \quad \kappa_{\log}^+(p) \ge \kappa_{\gamma\ell}^+(p),$$

for all $p \in \mathrm{int}(\Delta^2)$.

Rephrasing the previous discussion we have obtained the following characterization of fundamentality.

**Theorem 2.19.** *A loss function $\ell \in \mathcal{L}$ is fundamental if and only if there exist numbers $\eta, \gamma > 0$, such that for any $p \in \mathrm{int}(\Delta^2)$, there are translation vectors $x_p$ and $y_p$ in $\mathbb{R}_{\ge 0}^2$ such that*

$$\mathrm{spr}(\eta\ell + x_p) \subset \mathrm{spr}(\ell_{\log}) \subset \mathrm{spr}(\gamma\ell + y_p).$$

## 2.6 Constructing new mixable losses from previous

We now observe how mixability helps us to construct new proper, fair and mixable functions from previous proper, fair and mixable losses. We first define a family of losses that will serve to illustrate the idea. We set $\Phi = \Phi_{\mathrm{std}}$ and $\lambda = \ell_{\log}$. Let $a > 0$ and define the loss function $\lambda^a \colon \Delta^2 \longrightarrow \mathbb{R}_{\ge 0}^2$

$$\lambda^a(p) = a\lambda(p).$$

It can be readily checked that $\kappa_{\widetilde{\lambda^a}}(t) = a^{-1}\kappa_\lambda^+(t)$, thus since

$$\frac{\kappa_{\widetilde{\lambda^a}}(t)}{\kappa_\lambda^+(t)} = \frac{1}{a},$$

it follows that $\lambda^a$ is 1-mixable for $a \le 1$ and it is not if $a > 1$. Note that $\lambda^a$ is still proper and fair. Take then $a < 1$, we can readily see that there exists a proper, fair an mixable loss function $\lambda^*$ such that

$$\lambda = \lambda^a + \lambda^*.$$

Indeed, $\lambda^* = \lambda - \lambda^a = \lambda^{1-a}$, which is fair, proper and 1-mixable.

This process works in a more general setting than scalings of $\lambda$. Consider for example the spherical loss $\sigma$ defined in coordinates by

$$\widetilde{\sigma}(t) = \left(1 - \frac{t}{\sqrt{t^2+(1-t)^2}}, 1 + \frac{t}{\sqrt{t^2+(1-t)^2}}\right).$$

It can be easily checked that this is bounded, proper and fair and that $\kappa_{\widetilde{\sigma}}(t) = 1$. Thus

$$\frac{\kappa_{\widetilde{\sigma}}^+(t)}{\kappa_\lambda^+(t)} = \frac{(t^2+(1-t)^2)^{3/2}}{t(1-t)} > 1,$$

thus $\sigma$ is 1-mixable. Thus, as before, there is a loss function $\ell^*$ such that $\lambda = \sigma + \ell^*$. Moreover, the loss function given (in coordinates) by

$$\ell^*(t) = \lambda(t) - \sigma(t) = \left(-\ln(t) - 1 + \frac{t}{\sqrt{t^2+(1-t)^2}}, -\ln(1-t) - 1 - \frac{t}{\sqrt{t^2+(1-t)^2}}\right),$$

which can be seen to be unbounded, proper, fair and mixable.

We close this subsection with the following observation. Suppose that $\ell$ is a proper, fair, mixable loss function with mixability constant $\eta > 0$. Then the loss function $\ell^\eta = \eta\ell$ is 1-mixable. Thus, there exists a proper, fair, mixable loss $\ell^*$ such that

$$\ell_{\log} = \ell^\eta + \ell^*.$$

As we will see in Section 4, the previous observation can be interpreted from the point of view of the superprediction sets of the involved loss functions and convex geometry: $\mathrm{spr}(\eta\ell)$ *slides freely inside* $\mathrm{spr}(\lambda)$ (see Theorem 4.23).

## 2.7 Composite losses and the canonical link

In this part we discuss composite losses following (Reid & Williamson, 2010). Let us recall their setting. Let $\mathcal{V} \subset \mathbb{R}$ be a set of prediction values. A *link function* is a continuous map $\psi\colon [0,1] \longrightarrow \mathcal{V}$. Given a loss function $\widetilde{\varrho}\colon \{0,1\} \times [0,1] \longrightarrow \mathbb{R}$ and assuming $\mathcal{V} = \mathbb{R}$, if $\psi$ is invertible, we define the *composite loss* $\varrho^\psi$ as

$$\widetilde{\varrho}^\psi(y,v) = \widetilde{\varrho}(y, \psi^{-1}(v)).$$

**Definition 2.20.** *A composite loss $\widetilde{\varrho}^\psi$ is a* proper composite loss *if $\widetilde{\varrho}$ is a proper loss in the sense of Reid & Williamson (2010).*

Recall that in (Reid & Williamson, 2010), $\Phi = \Phi_{\mathrm{std}}$ is implicitly assumed. Then, given a loss function $\widetilde{\varrho}$ (in the (Reid & Williamson, 2010) sense), we can construct a loss function $\varrho\colon \Delta^2 \longrightarrow \mathbb{R}_{\geq 0}^2$, by $\varrho = \widetilde{\varrho} \circ \Phi_{\mathrm{std}}^{-1}$. Then, the composite loss $\widetilde{\varrho}^\psi$ can be expressed as

$$\begin{aligned}
\widetilde{\varrho}^\psi(v) &= (\widetilde{\varrho} \circ \psi^{-1})(v) \\
&= (\varrho \circ \Phi_{\mathrm{std}} \circ \psi^{-1})(p) \\
&= \left(\varrho \circ (\Phi_{\mathrm{std}} \circ \psi^{-1})\right)(p)
\end{aligned}$$

In other words, the composite loss $\widetilde{\varrho}^\psi$ is the local expression of $\varrho$ with respect to the parametrization $\Phi = \Phi_{\mathrm{std}} \circ \psi^{-1}$ of $\Delta^2$. We denote the local expression of $\varrho$ with respect to $\Phi$ by $\widehat{\varrho}$, that is $\widehat{\varrho} := \varrho \circ \Phi_{\mathrm{std}} \circ \Psi^{-1} = \widetilde{\varrho} \circ \Psi^{-1}$

To show how this reconciliation of terms work, we obtain a result similar to Corollary 12 in (Reid & Williamson, 2010). Suppose that a composite loss $\widetilde{\varrho}^\psi$ is given and it has differentiable partial losses (i.e., the corresponding

loss $\varrho$ is in $\mathcal{L}$), furthermore, we assume that $\psi$ is a diffeomorphism which in one dimension means it is strictly monotonic. Then we know that $\widetilde{\varrho}^{\psi}$ is strictly proper if and only if $\varrho$ is strictly proper (by definition). This implies that $p$ is normal to $\varrho(\Delta^2)$ at $\varrho(p)$ for all $p \in \text{int}(\Delta^2)$ and its curvature is positive (with respect to the unit normal pointing towards $\mathbb{R}^2_{\geq 0}$). This means for all $v \in \mathcal{V}$,

$$
\begin{aligned}
0 &= \langle \widehat{\varrho}'(v), \Phi(v) \rangle \\
&= \widehat{\varrho}'_1(v)\Phi_1(v) + \widehat{\varrho}'_2(v)\Phi_2(v) \\
&= \widetilde{\varrho}'_1(\psi^{-1}(v))(\psi^{-1})'(v)\Phi_1(v) + \widetilde{\varrho}'_2(\psi^{-1}(v))(\psi^{-1})'(v)\Phi_2(v) \\
&= \widetilde{\varrho}'_1(\psi^{-1}(v))\Phi_1(v) + \widetilde{\varrho}'_2(\psi^{-1}(v))(1 - \Phi_1(v)),
\end{aligned}
$$

where we have used that $\psi$ is a diffeomorphism and that $\Phi_1 + \Phi_2 = 1$ for all parametrizations $\Phi$ of $\Delta^2$. Therefore, we have

$$
\Phi_1(v)\left(\widetilde{\varrho}'_1(\psi^{-1}(v)) - \widetilde{\varrho}'_2(\psi^{-1}(v))\right) = -\widetilde{\varrho}'_2(\psi^{-1}(v)),
$$

that is

$$
\psi^{-1}(v) = \frac{\widetilde{\varrho}'_2(\psi^{-1}(v))}{\left(\widetilde{\varrho}'_2(\psi^{-1}(v)) - \widetilde{\varrho}'_1(\psi^{-1}(v))\right)}
$$

for all $v \in \mathcal{V}$.

Since we are working with valid reparametrizations the choice of $\Psi$ will not affect the curvature of $\varrho$. Hence we obtain

**Corollary 2.21.** *A composite loss $\widetilde{\varrho}^{\psi}$ is strictly proper if and only if $\varrho \in \mathcal{L}$ is strictly proper and $\psi$ satisfies*

$$
\psi^{-1}(v) = \frac{\widetilde{\varrho}'_2(\psi^{-1}(v))}{\left(\widetilde{\varrho}'_2(\psi^{-1}(v)) - \widetilde{\varrho}'_1(\psi^{-1}(v))\right)}
$$

*for all $v \in \mathbb{R}$.*

**Remark 2.22.** *We have seen that whether a loss function $\ell \in \mathcal{L}$ is strictly proper or not, depends on whether conditions (2.3) and (2.4) hold or not. Notice that under a (admissible) change of coordinates, for example given by a link $\psi$, (2.4) will not be modified. However, (2.3) might change (since in a way, we are changing the "velocity" at which we move on $\ell(\Delta^2)$). Hence, Corollary 2.21 is giving us a way to define the set of admissible links (or reparametrizations of $\Delta^2$) given a loss function $\ell$ and the standard parametrization of $\Delta^2$. In this case, the new parametrization is given by $\Phi = \Phi_{\text{std}} \circ \psi^{-1}$.*

For applications, it is desired to be able to work with a given composite loss $\widetilde{\varrho}^{\psi}$, and moreover, to have convexity of the partial losses $\widetilde{\varrho}^{\psi}_1$ and $\widetilde{\varrho}^{\psi}_2$. From our point of view, we see $\widetilde{\varrho}^{\psi}$ as the local expression of some $\varrho \colon \Delta^2 \longrightarrow \mathbb{R}$, so that $\widehat{\varrho} := \varrho \circ \Phi = \varrho \circ \left(\Phi_{\text{std}} \circ \psi^{-1}\right) = (\varrho \circ \Phi_{\text{std}}) \circ \psi^{-1} = \widetilde{\varrho} \circ \psi^{-1}$.

Let us work with the partial losses separately:

$$
\frac{d}{dv}\widehat{\varrho}_1(v) = \widetilde{\varrho}'_1(\psi^{-1}(v))(\psi^{-1})'(v)
$$

$$
\frac{d}{dv}\widehat{\varrho}_2(v) = \widetilde{\varrho}'_2(\psi^{-1}(v))(\psi^{-1})'(v)
$$

Proceeding as in the proof of Lemma 2.4, properness implies

$$
\begin{aligned}
0 &= \partial_v \widetilde{L}(u,v)|_{v=u} \\
&= \widetilde{\varrho}'_1(\psi^{-1}(v))(\psi^{-1})'(v)\Phi_1(u) + \widetilde{\varrho}'_2(\psi^{-1}(v))(\psi^{-1})'(v)\Phi_2(u)|_{s=u}
\end{aligned}
$$

or, equivalently,

$$0 = \widetilde{\varrho}_1'(\psi^{-1}(v))\Phi_1(v) + \widetilde{\varrho}_2'(\psi^{-1}(v))\Phi_2(v).$$

Therefore, we can define $w$ as

$$w(v) := w_{\widetilde{\varrho}}(\Psi^{-1}(v)) = \frac{\widetilde{\varrho}_2'(\psi^{-1}(v))}{\Phi_1(v)} = -\frac{\widetilde{\varrho}_1'(\psi^{-1}(v))}{\Phi_2(v)}, \tag{2.11}$$

where $w_{\widetilde{\varrho}}$ is the weight of $\widetilde{\varrho}$, we can rewrite the derivatives of the partial losses of $\widehat{\varrho}$ as

$$\frac{d\widehat{\varrho}_1}{dv}(v) = -w(v)\Phi_2(v)(\psi^{-1})'(v),$$

$$\frac{d\widehat{\varrho}_2}{dv}(v) = w(s)\Phi_1(v)(\psi^{-1})'(v).$$

Taking second derivatives we have

$$\frac{d^2\widehat{\varrho}_1}{dv^2}(v) = -\left[w(v)(\psi^{-1})'(v)\right]'\Phi_2(v) - \left[w(v)(\psi^{-1})'(v)\right]\Phi_2'(v),$$

$$\frac{d^2\widehat{\varrho}_2}{dv^2}(v) = \left[w(v)(\psi^{-1})'(v)\right]'\Phi_1(v) + \left[w(v)(\psi^{-1})'(v)\right]\Phi_1'(v).$$

A way to guarantee both expressions are positive is as follows. Assume w.l.o.g. that $(\psi^{-1})' > 0$. Since we are assuming $w > 0$, $\widehat{\varrho}_2$ is increasing and $\widehat{\varrho}_1$ is decreasing (also we have $\Phi_1$ is increasing and $\Phi_2$ is decreasing). We readily see that imposing

$$w(v)(\psi^{-1})'(v) = 1$$

for all $v \in \mathbb{R}$, is enough to guarantee both second derivatives to be strictly positive.

**Definition 2.23.** *Given $\varrho \in \mathcal{L}$ strictly proper, we define the* canonical link *$\psi$ as the link defined by*

$$(\psi^{-1})'(v) = \frac{\psi^{-1}(v)}{\widetilde{\varrho}_2'(\psi^{-1}(v))} = \frac{1}{w(v)}, \tag{2.12}$$

*for $v \in \mathcal{V}$, where $w$ is defined in* (2.11).

The differential equation (2.12) can be seen as separable ordinary differential equation, which is solvable for loss functions in $\mathcal{L}$.

To give a geometric meaning, we look at the norm of the velocity of the loss curve $\alpha(v) = \widehat{\varrho}(v)$.

$$|\alpha'(v)|^2 = w(v)^2(\psi^{-1})'(v)^2\left[\Phi_1(v)^2 + \Phi_2(v)^2\right]$$

By assuming $w(s)(\psi^{-1})'(s) = 1$ and $\Phi = \Psi$, we have

$$|\alpha'(s)|^2 = \left[\Phi_0(\psi^{-1}(s))^2 + \Phi_1(\psi^{-1}(s))^2\right].$$

Thus the canonical link gives a parametrization of $\Delta^2$ such that $\widehat{\varrho}$ is a curve such that its velocity vector at $v$ coincides with the length of the vector $\Phi(\psi^{-1}(v))$. In other words, it is a parametrization of the loss curve $\varrho(\text{int}(\Delta^2))$ such that for $\varrho(p) = \widehat{\varrho}(v) \in \ell(\text{int}(\Delta^2))$, the tangent vector at the point has length $|p|$. We close this discussion with a charcterization of the canonical link.

**Theorem 2.24.** *Let $\varrho \in \mathcal{L}$ be a strictly proper loss function and $\psi$ its canonical link. The reparametrization of $\varrho$ determined by its canonical link is a parametrization of $\varrho(\text{int}(\Delta^2))$ with weight equal to 1.*

*Proof.* Let $\widehat{\varrho} = \varrho \circ (\Phi_{\text{std}} \circ \psi^{-1}) = \widetilde{\varrho} \circ \psi^{-1}$ be the reparametrization of $\varrho(\text{int}(\Delta^2))$ determined by the canonical link. Since

$$\widehat{\varrho}_2'(v) = \widetilde{\varrho}_2'(\psi^{-1}(v))(\psi^{-1})'(v),$$

for all $v \in \mathcal{V}$, and from Definition 2.23

$$\widetilde{\varrho}_2'(\psi^{-1}(v))) = \frac{\psi^{-1}(v)}{(\psi^{-1})'(v)}, \tag{2.13}$$

for all $v \in \mathcal{V}$, we have

$$\widehat{\varrho}_2'(v) = \psi^{-1}(v).$$

Thus $w_{\widehat{\varrho}}(v) = \left| \frac{\widehat{\varrho}_2'(v)}{\Phi_1(v)} \right| = 1.$ $\qquad\square$

## 3 Mixability for Multi-Class Classification

Now we focus our attention on multi-class classification loss functions, that is, maps $\ell \colon \Delta^n \longrightarrow \mathbb{R}^n_{\geq 0}$ given by the partial losses

$$\ell(p) = (\ell_1(p), ..., \ell_n(p)).$$

Our main goal is to interpret mixability as a geometric comparison of a given loss function $\ell$ to the log loss, as we did for the binary case. As suggested by the comments after Remark 2.11, the extra work of characterizing properness and mixability in a geometric way (coordinate independent) will pay off since to carry out the comparison we will look at the *scalar second fundamental forms* of $\ell(\text{int}(\Delta^n))$ and $\ell_{\log}(\text{int}(\Delta^n))$. The scalar second fundamental form measures how a Riemannian manifold curves inside an "ambient space", in this case how $\ell(\text{int}(\Delta^n))$ curves inside $\mathbb{R}^n$ (see Appendix A for details).

The definition of $\mathcal{L}$ (Definition 2.1) can be extended to higher dimensions.

**Definition 3.1.** *An* admissible loss function *is a map* $\ell \colon \Delta^n \longrightarrow \mathbb{R}^n_{\geq 0}$ *such that*

(i) $\ell(\text{int}(\Delta^n)) \subset \mathbb{R}^n_{\geq 0}$ *is a* $(n-1)$-*manifold of class* $C^2$,

(ii) *there exists a* $C^1$ *map* $\mathbf{n} \colon \ell(\text{int}(\Delta^n)) \to N\ell(\text{int}(\Delta^n))$, $\mathbf{n}(\ell(p)) =: \mathbf{n}_{\ell(p)}$, *where* $N\ell(\text{int}((\Delta^n))$ *is the normal space of* $\ell(\text{int}((\Delta^n))$, *and*

(iii) $\mathbf{n}(\ell(p))$ *or* $-\mathbf{n}(\ell(p))$ *belongs to* $\mathbb{R}^n_{>0}$ *for all* $p \in \text{int}(\Delta^n)$.

*We denote the set of admissible loss functions as* $\mathcal{L}_n$, *or simply* $\mathcal{L}$ *when the dimension is clear from context.*

We fix the log loss and denote it for convenience by $\lambda := \ell_{\log} \colon \Delta^n \longrightarrow \mathbb{R}^n_{\geq 0}$, as the map

$$\lambda(p) = (-\ln(p_1), ..., -\ln(p_n)),$$

for $p = (p_1, ..., p_n) \in \Delta^n$.

Let $\ell \in \mathcal{L}_n$ and consider a parametrization $\Phi \colon D \subset \mathbb{R}^{n-1} \longrightarrow \Delta^n$ of $\Delta^n$ around $p \in \text{int}(\Delta^n)$. The local expression of the conditional risk (using the parametrization $\Phi \times \Phi$ of $\Delta^n \times \Delta^n$ around $(p, p)$) is given by

$$\widetilde{L}(t, s) = \langle \widetilde{\ell}(s), \Phi(t) \rangle = \sum_{k=1}^n \widetilde{\ell}_k(s)\Phi_k(t),$$

where $t = (t_1, ...t_{n-1}), s = (s_1, ..., s_{n-1}) \in D$ and $\widetilde{\ell} = \ell \circ \Phi$.

Imposing $\ell$ to be proper implies that when fixing $t$, $s = t$ is a critical point of $\widetilde{L}(t, \cdot)$, that is,

$$0 = \partial_{s_i} \widetilde{L}(t, \cdot)|_{s=t} = \langle \partial_{s_i} \widetilde{\ell}(t), \Phi(t) \rangle$$

for all $i \in \{1, ..., n-1\}$. Note that since the tangent space of $M_\ell$ at $\widetilde{\ell}(t)$, $T_{\widetilde{\ell}(t)} \widetilde{\ell}(U)$, is generated by $\{\partial_{s_1} \widetilde{\ell}(t), ..., \partial_{s_{n-1}} \widetilde{\ell}(t)\}$, we conclude that $\Phi(t)$ is a normal vector. In other words, as before, we have

$$\mathbf{n}(\ell(p)) = \pm \frac{p}{|p|},$$

for all $p \in \text{int}(\Delta^n)$.

The fact that $\widetilde{L}(t, \cdot)$ achieves a minimum at $s = t$ (at interior points) is equivalent to requiring that the Hessian, $D^2 \widetilde{L}$, is positive definite at $s = t$. The Hessian of $\widetilde{L}(t, \cdot)$ at $s = t$ is given by

$$[D^2 \widetilde{L}]_{ij}(t) = \partial_{s_j s_i} \widetilde{L}(t, \cdot)|_{s=t} = \langle \partial^2_{s_j s_i} \widetilde{\ell}(t), \Phi(t) \rangle.$$

The next step is to relate $[D^2 \widetilde{L}]_{ij}(t)$ to the scalar second fundamental form $h$ of $M_\ell = \ell(\Delta^n)$ (see Appendix A for its definition). More precisely, we compute the $h$ with respect to a local parametrization $\Phi$ of $\Delta^n$, i.e., we obtain the matrix $[h_{ij}]$ representing $h$. To do this we need to compute the second derivatives of its parametrization $\widetilde{\ell} = \ell \circ \Phi$ (Appendix A). Since,

$$\partial_{s_i} \widetilde{\ell}(s) = \left( \partial_{s_i} \widetilde{\ell}_1(s), ..., \partial_{s_i} \widetilde{\ell}_{n-1}(s) \right)$$

we have

$$\partial^2_{s_j s_i} \widetilde{\ell}(s) = \left( \partial^2_{s_j s_i} \widetilde{\ell}_1(s), ..., \partial^2_{s_j s_i} \widetilde{\ell}_{n-1}(s) \right).$$

The scalar second fundamental form (with respect to the normal vector pointing towards $\mathbb{R}^n_{\geq 0}$) is then given by

$$\begin{aligned} h_{ij}(s) = h(\partial_{s_i} \widetilde{\ell}(s), \partial_{s_j} \widetilde{\ell}(s)) &= \langle \partial^2_{s_j s_i} \widetilde{\ell}(s), \mathbf{n}(\widetilde{\ell}(s)) \rangle \\ &= \langle \partial^2_{s_j s_i} \widetilde{\ell}(s), \frac{\Phi(s)}{|\Phi(s)|} \rangle \\ &= \frac{1}{|\Phi(s)|} \langle \partial^2_{s_j s_i} \widetilde{\ell}(s), \Phi(s) \rangle \\ &= \frac{1}{|\Phi(s)|} [D^2 \widetilde{L}]_{ij}(s), \end{aligned} \tag{3.1}$$

for $i, j = 1, ..., n-1$, thus if $[D^2 \widetilde{L}]_{ij}(s)$ is positive definite, then the matrix $[h_{ij}](s)$ is positive definite. In this case its eigenvalues are strictly positive and hence, the *principal curvatures* of $M_\ell$ at $\ell(s)$ (see Appendix A), $\kappa_i^+(s)$ (with respect to the unit normal pointing towards $\mathbb{R}^n_{\geq 0}$) are all positive. Therefore, using a similar reasoning as we did in the case $n = 2$, we have obtained the following geometric characterization of properness (by following the same arguments as in Section 2).

**Lemma 3.2.** *Let $\ell \in \mathcal{L}_n$. $\ell$ is strictly proper if and only if $\mathbf{n}_\ell(p) = \pm p/|p|$ and the principal curvatures of $M_\ell$ at $\ell(p)$, $\kappa_i^+(p)$ $(i = 1, .., n-1)$, are strictly positive for all $p \in \text{int}(\Delta^n)$.*

We briefly explain how the comparison of scalar second fundamental forms will be performed. We follow a similar procedure as the one described in Section 2.4 for the case $n = 2$.

    1. We establish that given a proper loss function $\ell \in \mathcal{L}_n$, around every $p^* \in \text{int}(\Delta^n)$, $\ell(\text{int}(\Delta^n))$ can be parametrized as a graph of a function $f$ defined on a neighborhood around some $x^* \in \mathbb{R}^n$ such that $(x^*, f(x^*)) = \ell(p^*)$. We do this explicitly for the log loss $\lambda$.

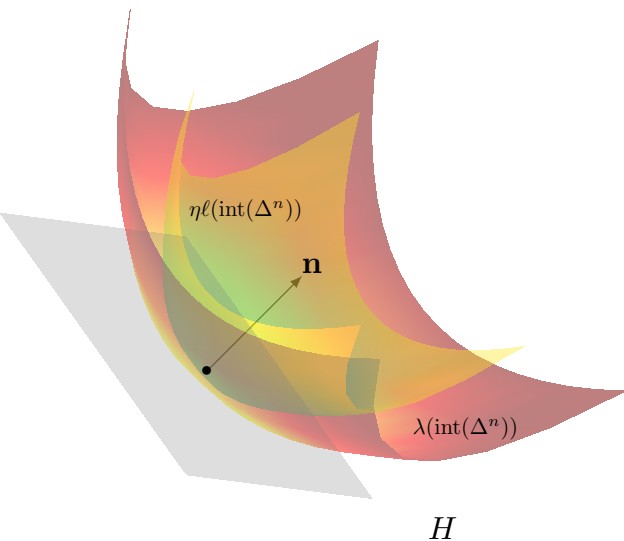

Figure 6: Geometric interpretation of $\eta$-mixability.

2. Since $\lambda$ and $\ell$ are proper, the normal vector to $\lambda(\text{int}(\Delta^n))$ and $\ell(\text{int}(\Delta^n))$ at $\lambda(p^*)$ and $\ell(p^*)$, respectively, is $p^*/|p^*|$. Hence we can identify their tangent spaces at these points. We do so and fix the parametrizations given in step (1).

3. By assuming $\eta$-mixability of $\ell$, we look at the principal curvatures of $E_\eta(\ell(\text{int}(\Delta^n))$ and prove an equivalent condition for them to be non-negative with respect to normal vector field pointing towards $E_\eta(\text{spr}(\ell))$ (i.e., convexity). The condition to be satisfied is seen to be comparison of the scalar second fundamental forms of $\lambda$ and $\ell$ that we can recognize by step (1).

4. We interpret this comparison as follows. Since the tangent spaces to $\ell(p^*)$ (and $\eta\ell(p^*)$) and $\lambda(p^*)$ coincide for the chosen point $p^*$, if we translate $\ell$ to coincide to $\lambda$ at $p^*$, call this tangent space $H$ (and note it can be indetified with the supporting plane of the loss functions at the given point). Then if we express (locally) $\eta\ell(\text{int}(\Delta^n))$ and $\lambda(\text{int}(\Delta^n))$ over $H$, the graph of $\eta\ell(\text{int}(\Delta^n))$ lies above the graph of $\lambda(\text{int}(\Delta^n))$. See Figure 6.

### 3.1 Representing proper loss functions as graphs over Euclidean spaces

When restricting to the set of admissible loss functions $\mathcal{L}_n$ ($n \geq 2$), we can represent losses as functions over $\mathbb{R}^{n-1}$ (a similar approach was taken in (van Erven et al., 2012); the difference relies on the fact that here we are after the comparison of second fundamental forms), which allows us to represent geometric quantities in a simple way. This will be useful to recognize these quantities when comparing a proper loss function $\ell$ to the log loss $\lambda$, as we did for the binary case in Section 2. Let $\ell \colon \Delta^n \longrightarrow \mathbb{R}^n_{\geq 0}$ be a proper loss in $\mathcal{L}_n$ given by

$$\ell(p) = (\ell_1(p), ..., \ell_n(p)).$$

Let $\Phi \colon \Delta^{n-1} \subset \mathbb{R}^{n-1} \longrightarrow \Delta^n$ be the standard parametrization of $\Delta^n$ given by

$$\Phi(s) = \Phi_{\text{std}}(s) = \left( s_1, ..., s_{n-1}, 1 - \sum_{i=1}^{n-1} s_i \right),$$

where $s = (s_1, ..., s_n) \in \Delta^{n-1}$. The local expression of $\widetilde{\ell}$ in these coordinates is then given by $\widetilde{\ell}(s) = (\ell \circ \Phi)(s)$, so that $\widetilde{\ell}_i(s) = (\ell_i \circ \Phi)(s)$. Also, we define the projection $\Pi \colon \mathbb{R}^n_{\geq 0} \longrightarrow \mathbb{R}^{n-1}_{\geq 0}$ as $\Pi(y_1, ..., y_n) = (y_1, ..., y_{n-1})$.

Recall that properness implies that the normal vector of $M_\ell = \ell(\Delta^n)$ at $\ell(p)$ can be chosen to be $|p|^{-1}p$, for $p \in \text{int}(\Delta^n)$. As a consequence, the normal vector is never parallel to the hyperplane $\{(x_1, .., x_n) \in$

$\mathbb{R}^n \,|\, x^n = 0\}$, so that around any point $\ell(p)$ with $p \in \text{int}(\Delta^n)$, $M_\ell$ can be written as a graph over $\mathbb{R}^n_{\geq 0} \times \{0\}$ (as regular as $M_\ell$ is). In general, the existence of this function is guaranteed by the implicit function theorem, however, in our case we can give an explicit description of it as follows. The function $\Pi|_{M_\ell}$ is a map with injective derivative, say around $\ell(q)$ for a fixed $q \in \text{int}(\Delta^n)$, therefore, the inverse function theorem ensures the existence (and differentiability) of a local inverse, which we can denote by $\Pi|_{M_\ell}^{-1}$. This inverse map can be seen as a local parametrization of $M_\ell$. Thus, the local expression of $\ell$ (viewed as a map from $\Delta^n$ to $M_\ell$), $\bar{\ell} \colon D_q \subset \mathbb{R}^{n-1} \longrightarrow U_{\ell(q)} \subset \mathbb{R}^{n-1}$ (where the latter are small neighborhoods around $\Phi^{-1}(q)$ and $\Pi(\ell(q))$ respectively) is given by

$$\bar{\ell}(s) = (\Pi \circ \ell \circ \Phi)(s) = (\Pi \circ \widetilde{\ell})(s) = \left( \widetilde{\ell}_1(s), ..., \widetilde{\ell}_{n-1}(s) \right).$$

This map is a diffeomorphism and its inverse $\bar{\ell}^{-1} \colon U_{\ell(q)} \longrightarrow D_q$, will be denoted by

$$\bar{\ell}^{-1}(x) = \left( \widetilde{\ell}_1^{-1}(x), ..., \widetilde{\ell}_{n-1}^{-1}(s) \right).$$

We warn the reader about this abuse of notation, $\widetilde{\ell}_i^{-1}(x)$ is not the inverse of $\widetilde{\ell}_i(s)$, it is a map satisfying

$$x_i = \widetilde{\ell}_i(s),$$
$$s_i = \widetilde{\ell}_i^{-1}(x),$$
$$(\bar{\ell} \circ \bar{\ell}^{-1})(x) = x,$$
$$(\bar{\ell}^{-1} \circ \bar{\ell})(s) = s.$$

We want to define $f \colon U_{\ell_q} \longrightarrow \mathbb{R}$ such that $\text{graph}(f) \subset M_\ell$. We see that setting $U_{\ell(q)} \subset \Pi(M_\ell)$, so that it contains $\Pi(\ell(q))$, we arrive to

$$f(x) = (\ell_n \circ \Phi \circ \bar{\ell}^{-1})(x) = \widetilde{\ell}_n(\bar{\ell}^{-1}(x)).$$

We have obtained the following result.

**Lemma 3.3.** *Let $\ell \in \mathcal{L}_n$ be a strictly proper loss. Let $q \in \text{int}(\Delta^n)$. Then there exists an open set $U \subset \mathbb{R}^{n-1}_{\geq 0} \times \{0\}$ and a function $f \colon U \longrightarrow \mathbb{R}_{\geq 0}$ such that $M_\ell$ admits the parametrization*

$$\Phi^f(x) = (x, f(x)),$$

*around $\ell(q)$.*

Let $\ell$ and $f$ be as in Lemma 3.3. The unit normal vector field (pointing towards $\mathbb{R}^n_{\geq 0}$) is then given by

$$\mathbf{n}^\ell(x) = \frac{1}{\sqrt{|Df(x)|^2 + 1}} \left( -Df(x), 1 \right). \tag{3.2}$$

We proceed to calculate the scalar second fundamental form. The first and second derivatives of $\Phi^f$ are given by

$$\partial_k \Phi^f(x) = (e_k, \partial_k f(x)),$$
$$\partial_{km} \Phi^f(x) = (\mathbf{0}, \partial_{km} f(x)),$$

for $k, m = 1, ..., n-1$, where $e_k$ denotes the canonical basis of $\mathbb{R}^{n-1}$ and $\mathbf{0}$ is the $0$ vector of $\mathbb{R}^{n-1}$. Denote by $h^\ell$ the scalar second fundamental form of $M_\ell$. Thus with respect to this coordinates we have

$$h^\ell_{km}(x) = \langle \partial_{km} \Phi^f(x), \mathbf{n}^\ell(x) \rangle = \frac{1}{\sqrt{|Df(x)|^2 + 1}} \partial_{km} f(x), \tag{3.3}$$

for $k, m = 1, ..., n-1$.

### 3.1.1 $M_\lambda$ as a graph

Fix an arbitrary point $q^* \in \mathrm{int}(\Delta^n)$. The local expression of $\lambda$ (with respect to the standard parametrization $\Phi = \Phi_{\mathrm{std}}$ around $q^*$ and $\Pi$ around $\ell(q^*)$) is given by

$$\overline{\lambda}(s) = (-\ln(s_1), ..., -\ln(s_{n-1})),$$

thus, we have

$$\overline{\lambda}^{-1}(x) = \left(e^{-x_1}, ..., e^{-x_{n-1}}\right).$$

Fix $s^* = \Phi^{-1}(q^*)$. Thus, around $x^* = \Pi(\lambda(q^*))$, using Lemma 3.3, $M_\lambda$ around $\ell(q)$ can be described as

$$\Phi^g(x) = (x, g(x)).$$

Moreover, in this case we have the explicit expression $g(x) = -\ln(1 - \sum_{i=1}^{n-1} e^{-x_i})$. Notice that $\overline{\lambda}^{-1}(x^*) = s^*$. We now compute the scalar second fundamental form $h^\lambda$ of $\lambda$ at $x^*$.

$$\partial_k \Phi^g(x) = \left(e_k, -\frac{e^{-x_k}}{1 - \sum_{i=1}^{n-1} e^{-x_i}}\right),$$

$$\partial_{km} \Phi^g(x) = \left(\mathbf{0}, \frac{\delta_{km} e^{-x_k}}{1 - \sum_{i=1}^{n-1} e^{-x_i}} + \frac{e^{-x_k} e^{-x_m}}{\left(1 - \sum_{i=1}^{n} e^{-x_i}\right)^2}\right),$$

for $k, m = 1, ..., n-1$ (here $\delta_{km}$ denotes the *Kronecker delta*). In particular,

$$\partial_k \Phi^g(x^*) = \left(e_k, -\frac{s_k^*}{1 - \sum_{i=1}^{n-1} s_i^*}\right),$$

$$\partial_{km} \Phi^g(x^*) = \left(\mathbf{0}, \frac{\delta_{km} s_k^*}{1 - \sum_{i=1}^{n-1} s_i^*} + \frac{s_k^* s_m^*}{\left(1 - \sum_{i=1}^{n-1} s_i^*\right)^2}\right),$$

for $k, m = 1, ..., n-1$, and since $\mathbf{n}((x^*, g(x^*)) = \frac{1}{\sqrt{\sum_{i=1}^{n-1}(s_i^*)^2 + (1 - \sum_{i=1}^{n-1} s_i^*)^2}}(s^*, 1 - \sum_{i=1}^{n-1} s_i^*)$ we have

$$h_{km}^\lambda(x^*) = \langle \partial_{km} \Phi^g(x^*), \mathbf{n}((x^*, g(x^*))) \rangle \tag{3.4}$$

$$= \frac{1}{\sqrt{\sum_{i=1}^{n}(s_i^*)^2 + (1 - \sum_{i=1}^{n} s_i^*)^2}} \left(\delta_{km} s_k^* + \frac{s_k^* s_m^*}{1 - \sum_{i=1}^{n} s_i^*}\right), \tag{3.5}$$

for $k, m = 1, ..., n-1$.

**Remark 3.4.** *Note that if instead of $\lambda$ we would have used a translation of it, that is, for $c \in \mathbb{R}^n$, define a loss function $\varphi \colon \Delta^n \longrightarrow \mathbb{R}_{\geq 0}^n$ by*

$$\varphi(p) = \lambda(p) + c,$$

*we can repeat the previous computation. The only difference is that we would have a different point $x_*^c$ instead of $x_*$.*

### 3.2 Geometric interpretation of mixability

Mixability is defined as a property of the superprediction set of a proper loss $\ell \in \mathcal{L}_n$. More precisely, $\ell$ is mixable if and only if $E_\eta(\mathrm{spr}(\ell))$ is convex for some $\eta > 0$. As before, we can determine whether $E_\eta(\mathrm{spr}(\ell))$ is convex by looking at its boundary $\partial E_\eta(\mathrm{spr}(\ell)) = E_\eta(\ell(\Delta^n))$. $E_\eta(\mathrm{spr}(\ell))$ is convex if the principal curvatures of $E_\eta(\ell(\Delta^n))$ are non-negative (when defined with respect to the inner pointing normal vector) at all points. Since convexity is a global property that can be tested "locally everywhere", it makes sense to make the following definition.

**Definition 3.5** ($\eta$-Mixability at $p \in \Delta^n$). *We say that $\ell \in \mathcal{L}_n$ is $\eta$-mixable at $p \in \text{int}(\Delta^n)$ if $E_\eta(M_\ell)$ has non-negative principal curvatures with respect to the unit normal vector pointing towards $E_\eta(\text{spr}(\ell))$ at $E_\eta(\ell(p))$.*

Clearly, $\ell \in \mathcal{L}_n$ is $\eta$-mixable at all $p \in \text{int}(\Delta^n)$ if and only if it is $\eta$-mixable.

Let $\ell, \varrho \in \mathcal{L}_n$ be strictly proper. First, we note that properness implies that the second fundamental forms of $\ell$ and $\varrho$ can be compared in the following sense. Given $q^* \in \Delta^n$, note that the normal vector to $M_\ell$ and $M_\varrho$ can be chosen to be $q^*/|q^*|$. A translation does not affect the geometric properties of $M_\varrho$ (since it is an isometry of $\mathbb{R}^n$), thus we consider the translated loss $\varrho^{\ell(q^*)} \colon \Delta^n \longrightarrow \mathbb{R}^n$, given by

$$\varrho^{\ell(q^*)}(p) = \varrho(p) + [\ell(q^*) - \varrho(q^*)],$$

i.e., we translate $\varrho$ by the vector $c^{q^*} = \lambda(q^*) - \ell(q^*)$ so that both $\varrho^{q^*}$ and $\ell$ coincide when evaluated at $q^*$. Doing so allows us to identify the tangent spaces to $M_{\varrho^{\ell(q^*)}}$ and $M_\ell$ at $\varrho^{\ell(q^*)}(q^*) = \ell(q^*)$. We will call $\varrho^{\ell(q^*)}$ the *translation of $\varrho$ to $\ell(q^*)$*.

**Lemma 3.6.** *Let $\ell \in \mathcal{L}_n$ be strictly proper. Let $h^\ell$ and $h^\lambda$ denote the scalar second fundamental form of $M_\ell$ and $M_\lambda$ (the log loss), respectively. Then, $\ell$ is $\eta$-mixable at $p \in \text{int}(\Delta^n)$ if and only if*

$$h^\ell(\ell(p)) - \eta h^\lambda(\lambda(p)) \tag{3.6}$$

*is positive semi-definite, where $h^\ell$ and $h^\lambda$ denote the second fundamental forms of $\ell$ and $\lambda$ in the graphical coordintes described in Lemma 3.3. And therefore, $\ell$ is $\eta$-mixable if and only if (3.6) holds for all $p \in \text{int}(\Delta^n)$.*

*Proof.* Let $\ell \colon \Delta^n \longrightarrow \mathbb{R}^n_{\geq 0}$ be an admissible proper loss

$$\ell(p) = (\ell_1(p), ..., \ell_n(p)).$$

The $\eta$-exponential projection map $E_\eta \colon \mathbb{R}^n \longrightarrow \mathbb{R}^n$ is given by

$$E_\eta(y) = (e^{-\eta y_1}, ..., e^{-\eta y_n}).$$

Let $q^* \in \text{int}(\Delta^n)$ and write $M_\ell$ around $\ell(q^*)$ as the graph of a function $f$ over $\mathbb{R}^{n-1}$, defined on an open set $U^f_{x^*}$ containing $x^*$, such that $f(x^*) = \ell(q^*)$. We can directly give a parametrization of $E_\eta(M_\ell)$ around $E_\eta(\ell(q^*)) = E_\eta((x^*, f(x^*)))$ by

$$\Psi(x) = \left(e^{-\eta x_1}, ..., e^{-\eta x_{n-1}}, e^{-\eta f(x)}\right).$$

We proceed to compute the second fundamental form of $E_\eta(M_\ell)$ around $E_\eta(\ell(q^*))$ (with respect to the inward pointing unit normal vector). The first and second derivatives of $\Psi$ are given by

$$\partial_k \Psi(x) = \left(-\eta e^{-\eta x_k} e_k, -\eta \partial_k f(x) e^{-\eta f(x)}\right)$$

$$\partial_{km} \Psi(x) = \left(\eta^2 \delta_{km} e^{-\eta x_k} e_k, -\eta \partial_{km} f(x) e^{-\eta f(x)} + \eta^2 \partial_k f(x) \partial_m f(x) e^{-\eta f(x)}\right)$$

and noting that the (inward pointing) unit vector field is given by

$$\mathbf{n}(E_\eta(\ell(x))) = \frac{1}{\left(\sum_{i=1}^{n-1} \partial_i f(x)^2 e^{2\eta x_i} + e^{2\eta f(x)}\right)^{1/2}} \left(\partial_1 f(x) e^{\eta x_1}, ..., \partial_n f(x) e^{\eta x_n}, -e^{\eta f(x)}\right)$$

Therefore, letting $\mathcal{E}^\eta := \mathcal{E}^\eta(U_f) = E_\eta(f(U_f))$, the second fundamental form of $\mathcal{E}^\eta$ at $E_\eta((x^*, f(x^*)))$ is given by

$$h^{\mathcal{E}^\eta}_{km}(x^*)$$
$$= \langle \partial_{km} \Psi(x^*), \mathbf{n}(\ell^\eta(x^*)) \rangle$$

$$= \frac{1}{\left(\sum_{i=1}^{n-1} \partial_i f(x^*)^2 e^{2\eta x_i^*} + e^{2\eta f(x^*)}\right)^{1/2}} \left[ \langle \eta^2 \delta_{km} e^{-\eta x_k^*} e_k, \sum_{i=1}^{n} \partial_i f(x^*) e^{\eta x_i^*} e_i \rangle_{\mathbb{R}^{n-1}} \right.$$

$$\left. + \eta \partial_{km} f(x^*) - \eta^2 \partial_k f(x^*) \partial_m f(x^*) \right]$$

$$= \frac{\eta}{\left(\sum_{i=1}^{n-1} \partial_i f(x^*)^2 e^{2\eta x_i^*} + e^{2\eta f(x^*)}\right)^{1/2}} \left[ \eta \delta_{km} \partial_k f(x^*) + \partial_{km} f(x^*) - \eta \partial_k f(x^*) \partial_m f(x^*) \right].$$

Thus, since the convexity of $E_\eta(\mathrm{spr}(\ell))$ is equivalent to the principal curvatures of $E_\eta(M_\ell)$ being non-negative at $q^*$ for all $q^* \in \mathrm{int}(\Delta^n)$ (with respect to the inner pointing normal vector), we see this will be the case if and only if the matrix

$$A_{km} = \partial_{km} f(x^*) - \eta \left[ -\delta_{km} \partial_k f(x^*) + \partial_k f(x^*) \partial_m f(x^*) \right]$$

is positive semi-definite for all $x^*$ corresponding to $q^* \in \mathrm{int}(\Delta^n)$.

Note that since we have a graphical parametrization $\Phi^f$ of $M_\ell$ around $x^* \in U$, we have

$$\partial_k \Phi^f(x^*) = (e_k, \partial_k f(x^*))$$

and by (3.2),

$$\mathbf{n}(x^*, f(x^*)) = \frac{1}{\sqrt{|Df(x^*)|^2 + 1}} \left( -Df(x^*), 1 \right).$$

On the other hand, since the normal vector to $\Phi^f(U)$ at $(x^*, f(x^*))$ is $\frac{q^*}{|q^*|}$, we have

$$\mathbf{n}((x^*, f(x^*))) = \frac{1}{\sqrt{\sum_{i=1}^{n}(s_i^*)^2 + (1 - \sum_{i=1}^{n} s_i^*)^2}} \left( s_1^*, \ldots, s_n^*, 1 - \sum_{i=1}^{n-1} s_i^* \right),$$

for $s^* \in \mathbb{R}^{n-1}$ such that $\Phi(s^*) = q^*$.

By properness we know that

$$0 = \langle \partial_k \Phi^f(x^*), \mathbf{n}((x^*, f(x^*))) \rangle$$

$$= \frac{1}{\sqrt{\sum_{i=1}^{n}(s_i^*)^2 + (1 - \sum_{i=1}^{n} s_i^*)^2}} \left[ s_k^* + \partial_k f(x^*) \left( 1 - \sum_{i=1}^{n-1} s_i^* \right) \right]$$

thus

$$\partial_k f(x^*) = \frac{-s_k^*}{1 - \sum_{i=1}^{n-1} s_i^*},$$

and also

$$1 + |Df(x^*)|^2 = 1 + \frac{\sum_{j=1}^{n-1}(s_j^*)^2}{\left(1 - \sum_{i=1}^{n-1} s_i^*\right)^2} = \frac{\sum_{j=1}^{n-1}(s_j^*)^2 + \left(1 - \sum_{i=1}^{n-1} s_i^*\right)^2}{\left(1 - \sum_{i=1}^{n-1} s_i^*\right)^2}. \tag{3.7}$$

Using (3.3) and the previous observations, we can rewrite the terms of $A_{km}$ as

$$\partial_{km} f(x^*) = \frac{\sqrt{|Df(x^*)|^2 + 1}}{\sqrt{|Df(x^*)|^2 + 1}} \partial_{km} f(x^*)$$

$$= \frac{1}{\left(1 - \sum_{i=1}^{n-1} s_i^*\right)} \sqrt{\sum_{j=1}^{n-1} (s_j^*)^2 + \left(1 - \sum_{i=1}^{n-1} s_i^*\right)^2} \, h_{km}^{\ell}(x^*)$$

and

$$[-\delta_{km} \partial_k f(x^*) + \partial_k f(x^*) \partial_m f(x^*)] = \frac{\delta_{km} s_k^*}{1 - \sum_{i=1}^{n-1} s_i^*} + \frac{s_k^* s_m^*}{\left(1 - \sum_{i=1}^{n-1} s_i^*\right)^2}.$$

Now, consider the log loss $\lambda$ and its translation to $\ell(q^*)$ which we denote by $\lambda^*$ to simplify the notation. That is, we have

$$\lambda^* = \lambda(p) + [\ell(q^*) - \lambda(q^*)].$$

As discussed in Remark 3.4, we can write $M_{\lambda^*}$ as a graph around $x^*$ (since $\lambda^*(q^*) = \ell(q^*)$). The scalar second fundamental form of $M_{\lambda^*}$ at $\lambda^*(q^*)$ is then given by

$$h_{ij}^{\lambda^*}(x^*) = \frac{1}{\sqrt{\sum_{i=1}^{n} (s_i^*)^2 + (1 - \sum_{i=1}^{n} s_i^*)^2}} \left( \delta_{km} s_k^* + \frac{s_k^* s_m^*}{1 - \sum_{i=1}^{n} s_i^*} \right). \tag{3.8}$$

This readily implies that

$$[-\delta_{km} \partial_k f(x^*) + \partial_k f(x^*) \partial_m f(x^*)] = \frac{1}{\left(1 - \sum_{i=1}^{n-1} s_i^*\right)} \sqrt{\sum_{j=1}^{n-1} (s_j^*)^2 + \left(1 - \sum_{i=1}^{n-1} s_i^*\right)^2} \, h_{ij}^{\lambda^*}(x^*).$$

Therefore, $\ell$ is $\eta$-mixable at $q^*$ if and only if we have that

$$[h_{ij}^{\ell}](x^*) - \eta[h_{ij}^{\lambda^*}](x^*)$$

is semi-positive definite. Since $q^*$ was arbitrary the result follows. $\qquad\square$

**Remark 3.7.** *The previous comparison of second fundamental forms is possible because properness forces the induced metrics by $\ell$ and $\lambda$ to coincide at $\ell(q^*) = \lambda^*(q^*)$, that is, $[g_{ij}^{\ell}](x^*) = [g_{ij}^{\lambda^*}](x^*)$ (see Appendix A and Remark A.3). The conclusion of Theorem 3.6 does not necessarily hold if one takes a different coordinate system.*

In order to get a geometric interpretation (i.e., independent of coordinates) we note the following:

$$0 \leq [h_{ij}^{\ell}](x^*) - \eta[h_{ij}^{\lambda^*}](x^*)$$
$$= [h_{ij}^{\ell}](x^*)[g_{ij}^{\ell}]^{-1}(x^*)[g_{ij}^{\ell}](x^*) - \eta[h_{ij}^{\lambda^*}](x^*)[g_{ij}^{\lambda^*}]^{-1}(x^*)[g_{ij}^{\lambda^*}](x^*)$$
$$= \left( [h_{ij}^{\ell}](x^*)[g_{ij}^{\ell}]^{-1}(x^*) - \eta[h_{ij}^{\lambda^*}](x^*)[g_{ij}^{\lambda^*}]^{-1}(x^*) \right) [g_{ij}^{\ell}](x^*).$$

The matrices $[h_{ij}^{\ell}](x^*)[g^{\ell}]^{-1}(x^*)$ and $[h_{ij}^{\lambda^*}](x^*)[g^{\lambda^*}]^{-1}(x^*)$ are the local expression of the Weingarten map (see (Lee, 2018) for its definition and properties) of $\ell$ and $\lambda$ respectively. The eigenvalues of these matrices are the principal curvatures of $M_{\ell}$ and $M_{\lambda}$ (and they are independent of coordinates), and the determinants are their Gaussian curvatures. From here it also follows that

$$\eta \left[ \frac{1}{\eta} [h_{ij}^{\ell}](x^*)[g_{ij}^{\ell}]^{-1}(x^*) - [h_{ij}^{\lambda^*}](x^*)[g_{ij}^{\lambda^*}]^{-1}(x^*) \right] [g_{ij}^{\ell}](x^*)$$
$$= \eta \left[ [h_{ij}^{\eta\ell}](x^*)[g_{ij}^{\eta\ell}]^{-1}(x^*) - [h_{ij}^{\lambda^*}](x^*)[g_{ij}^{\lambda^*}]^{-1}(x^*) \right] [g_{ij}^{\ell}](x^*),$$

that is,

$$[W_{ij}^{\eta\ell}] - [W_{ij}^{\lambda}] \geq 0, \tag{3.9}$$

where $W^{\ell}$ denotes the Weingarten map of the loss function $\ell$. Then once a system of coordinates around $p \in \Delta^n$ is chosen the relation (3.9) holds. A priori, the relation obtained between the Weingarten maps of $\ell$ and $\lambda$ does not provide much information, but it does points to look at the loss function $\eta\ell$. With this in mind Lemma 3.6 does give a direct geometric interpretation as follows. Let $\ell \colon \Delta^n \longrightarrow \mathbb{R}_{\geq 0}^n$ in $\mathcal{L}$ be a proper loss. Given a point $q \in \Delta^n$ we know that around $\ell(q)$, $M_{\ell}$ can be parametrized with $\Phi^{\bar{f}}(x) = (x, f(x))$ for some function $f$ around the point $\Pi(\ell(q))$. Let $x^* = \Phi(\ell(q))$. Consider now the proper loss $\varrho = \eta\ell$, for some $\eta > 0$. We readily see that $\varrho$ can be parametrized as $\Phi^g(y) = (y, g(y))$ with

$$g(y) = \eta f(\eta^{-1}y),$$

with $g$ defined around $y_q = \eta x^*$. Now we compute the second fundamental form of $\varrho$ at $y_q$. Notice that

$$\partial_i g(y)|_{y=\eta x^*} = \eta \partial_i f(\eta^{-1}x)|_{y=\eta x^*}\eta^{-1} = \partial_i f(x^*),$$
$$\partial_{ij} g(y)|_{y=\eta x^*} = \partial_{ij} f(\eta^{-1}x)|_{y=\eta x^*}\eta^{-1} = \eta^{-1}\partial_{ij} f(x^*),$$

and hence,

$$h_{ij}^{\varrho}(\eta x^*) = h_{ij}^{\eta\ell}(\eta x^*) = \eta^{-1}h^{\ell}(x^*).$$

Then assuming the hypothesis of Lemma 3.6, we obtain

$$h_{ij}^{\eta\ell}(\eta x^*) - h_{ij}^{\lambda}(x^*) = \eta^{-1}h_{ij}^{\ell}(x^*) - h_{ij}^{\lambda}(x^*) = \eta^{-1}\left(h^{\ell}(x^*) - \eta h_{ij}^{\lambda}(x^*)\right) \geq 0. \tag{3.10}$$

The supporting planes at $\eta\ell(p)$ and $\lambda(p)$ of $M_{\eta\ell}$ (or more precisely, of its translation to $\lambda(p)$) and $M_{\lambda}$, respectively, coincide (since the normal vectors are the same), we denote it by $H_p$. By looking at $M_{\eta\ell}$ and $M_{\lambda}$ locally as graphs over $H_p$, Lemma 3.6 gives the following comparison of graphs, which in turn can be regarded as local embeddability in the sense of convex geometry (see Definition 4.16 below).

**Theorem 3.8.** $\ell \in \mathcal{L}_n$ *proper is $\eta$-mixable if and only if for all $p \in \text{int}(\Delta^n)$ the local graph of the translation of $\eta\ell$ to $\lambda(p)$ over the supporting plane to both $M_{\ell_p}$ and $M_{\lambda}$ at $\lambda(p)$, $H_p$, lies above the graph of $\lambda$ over $H_p$.*

**Remark 3.9.** *Observe the resemblance of Lemma 3.6 to Theorem 10 in (van Erven et al., 2012). To recover the latter from our point of view we will first reinterpret Lemma 3.6 and Theorem 3.8 from a convex geometry point of view which will lead to a nice bridge between Lemma 3.6 and (van Erven et al., 2012, Theorem 10).*

## 4 Connections to convex geometry

In this section we reinterpret our results from the point of view of convex geometry. With this interpretation we can relate Theorem 3.8 to results in (van Erven et al., 2012) and (Williamson & Cranko, 2022). We first provide some background and state relevant results from convex geometry which are well-known and can be found in (Schneider, 2014) and can be adapted to our setting.

Let $K \subset \mathbb{R}^n$ be a convex set, that is

$$\lambda x + (1 - \lambda)y \in K$$

for all $x, y \in K$ and $\lambda \in [0, 1]$.

We define the *recession cone* of $K$ as the set

$$\text{rec}(K) = \{x \in \mathbb{R}^n : K + x \subset K\}.$$

The boundary of $K$ is denoted by $\partial K$, as since we will assume that $\partial K$ is a differentiable manifold we denote the interior (as a manifold) of $\partial K$ by $\text{int}(\partial K)$. As usual the *scaling* of $K$ by $\eta > 0$ and the *Minkowski sum* of $K$ and $L$ are defined as

$$\eta K = \{\eta k \in \mathbb{R}^n : k \in K\}, \tag{4.1}$$
$$K + L = \{k + l \in \mathbb{R}^n : k \in K, l \in L\}. \tag{4.2}$$

**Definition 4.1.** *Let $K$ be a closed convex set in $\mathbb{R}^n$. The* support function *of $K$, $\sigma(K,u)\colon \mathbb{R}^n \longrightarrow \overline{\mathbb{R}}$, is defined as*

$$\sigma(K,u) = \sup_{x \in K} \langle x, u \rangle.$$

*We sometimes denote it as $\sigma_K(u) \coloneqq \sigma(K,u)$.*

From the definition we know that

$$y \in K \iff \langle y, u \rangle \le \sigma_K(u) \text{ for all } u \in \mathbb{R}^n.$$

From (Schneider, 2014, Section 1.7) we have the following.

**Lemma 4.2** (Properties of $\sigma$). *Let $L, K \subset \mathbb{R}^n$ be closed convex sets.*

1. *$\sigma_L \le \sigma_K$ if and only if $L \subset K$.*

2. *$\sigma(K+t, u) = \sigma(K,u) + \langle t, u \rangle$ for all $t \in \mathbb{R}^n$.*

3. *$\sigma(K+L, u) = \sigma(K,u) + \sigma(L,u)$.*

**Definition 4.3.** *A function $f\colon D \subset \mathbb{R}^n \longrightarrow \overline{\mathbb{R}}$ is* convex *if its extension to $\mathbb{R}^n$ given by*

$$\widetilde{f}(x) = \begin{cases} f(x), & \text{if } x \in D \\ \infty, & \text{if } x \notin D \end{cases}$$

*is convex.*

The following lemma is a well-known result (see (Schneider, 2014, Theorem 1.7.1) for example).

**Lemma 4.4.** *Let $f\colon \mathbb{R}^n \to \mathbb{R}$ convex, closed and positively homogeneous, then $f$ is the support function of the convex, closed set*

$$K^f = \{x \in \mathbb{R}^n \mid \langle x, u \rangle \le f(u) \text{ for all } u \in \mathbb{R}^n\}.$$

**Definition 4.5.** *Let $L, K \subset \mathbb{R}^n$ and closed and convex. We say that $L$ is a* summand *of $K$ if there exists a convex, closed set $M \subset \mathbb{R}^n$ such that $K = M + L$.*

We will be mainly interested in sets $K$ whose recession cone is $\mathbb{R}^n_{\ge 0}$, hence we denote by $\mathcal{K}^n_*$ the set of closed, convex sets whose recession cone is $\mathbb{R}^n_{\ge 0}$. In the following we extend some common results in convex geometry which are usually stated for closed, compact convex sets in $\mathbb{R}^n$ (Schneider, 2014), however, some of them are easily extended to $\mathcal{K}^n_*$ (Shveidel, 2001).

**Lemma 4.6** (Basic properties of sets in $\mathcal{K}^n_*$). *Let $K, L \in \mathcal{K}^n_*$ and $\eta > 0$. Then, the following holds:*

*(1) $\eta K \in \mathcal{K}^n_*$,*

*(2) $\operatorname{rec}(K+L) = \mathbb{R}^n_{\ge 0}$,*

*(3) $K+L$ is closed, and*

*(4) $K+L \in \mathcal{K}^n_*$.*

*Proof.* In order to show (1), we need to show that $\eta K$ is closed, convex and $\operatorname{rec}(\eta K) = \mathbb{R}^n_{\ge 0}$. Let $x, y \in \eta K$ and $\lambda \in [0,1]$, then we have

$$\lambda x + (1-\lambda)y = \eta(\lambda k_x + (1-\lambda)k_y)$$

where $x = \eta k_x$ and $y = \eta k_y$ for some $k_x, k_y \in K$. Since $K$ is convex, then $\lambda k_x + (1-\lambda)k_y \in K$ and hence $\eta K$ is convex. Let $x_n \in K$ be a convergent sequence that converges to $x$. Then, there exists $k_{x_n} \in K$ such

that $x_n = \eta k_{x_n}$. Since $\eta$ is a constant, $\{k_{x_n}\}$ converges to $k_{x_\infty} \in K$ (since $K$ is closed). By the uniqueness of the limit, $x = \eta x_\infty \in \eta K$. Now, let $x \in \mathbb{R}^n_{\geq 0}$, we want to show that $\eta K + x \subset \eta K$. Take any $k \in K$,

$$\eta k + x = \eta \left( k + \frac{1}{\eta} x \right) \in \eta K$$

since $\frac{1}{\eta} x \in \mathbb{R}^n_{\geq 0}$. Conversely, if $x \in \text{rec}(\eta K)$, then for any $k_1 \in K$, we have

$$\eta k_1 + x \in \eta K$$

then there exists $k_2 \in K$, such that $\eta k_1 + x = \eta k_2$. Hence

$$k_1 + \frac{1}{\eta} x = k_2,$$

thus $\frac{1}{\eta} x \in \text{rec}(K) = \mathbb{R}^n_{\geq 0}$, thus $x \in \mathbb{R}^n_{\geq 0}$.

To show (2), let $x \in \mathbb{R}^n_{\geq 0}$. We want to show that $K + L + x \subset K + L$. Let $k \in K$ and $l \in L$, then

$$k + l + x \in K + L,$$

since $l + x \in L$. Thus $\mathbb{R}^n_{\geq 0} \subset \text{rec}(K + L)$. Now, suppose that there is $x \in \text{rec}(K + L)$ such that $x \notin \mathbb{R}^n_{\geq 0}$. Since $\text{rec}(K + L)$ is a cone, for all $\lambda > 0$, we have $\lambda x \in \text{rec}(K + L)$. Let $k \in K$ and $l \in L$. Then

$$k + l + \lambda x \in K + L \subset K.$$

Thus $\ell + \lambda x \in \text{rec}(K) = \mathbb{R}^n_{\geq 0}$ for all $\lambda > 0$, but notice that this is a contradiction since by picking $\lambda$ sufficiently large, $l + \lambda x \notin \mathbb{R}^n_{\geq 0}$. Thus $\text{rec}(K + L) = \mathbb{R}^n_{\geq 0}$.

For (3), see (Rockafellar, 1970, Theorem 8.2) and (Shveidel, 2001, Theorem 3.1). Finally, (4) is simply the combination of (2) and (3) (and the fact that $K + L$ is convex). $\qquad\square$

We now specialize the discussion to a particular type of sets $K \in \mathcal{K}^n_*$. First, suppose that the boundary $\partial K$ is of class $C^2$, then at each point $x \in \text{int}(\partial K)$ there is an *outward* pointing normal vector $\mathbf{u}_K(x)$. Thus, clearly, we can define a map $\mathbf{u}_K \colon \text{int}(\partial K) \longrightarrow \mathbb{S}^{n-1}$ assigning $u_K(x)$ to $x \in \text{int}(\partial K)$. We define

$$\mathbb{R}^n_{\leq 0} = \{ x \in \mathbb{R}^n \,:\, x = (x_1, ..., x_n), \text{ with } x_i \leq 0 \text{ for } i = 1, ..., n \},$$

so that

$$\text{int}(\mathbb{R}^n_{\leq 0}) = \{ x \in \mathbb{R}^n \,:\, x = (x_1, ..., x_n), \text{ with } x_i < 0 \text{ for } i = 1, ..., n \} = \mathbb{R}^n_{< 0}.$$

**Definition 4.7.** *Define $C^2_+(\mathcal{K}^n_*)$ as the collection of sets $K \in \mathcal{K}^n_*$ with boundary $\partial K$ of class $C^2$, and such that the map $\mathbf{u}_K$ is a $C^1$-diffeomorphism from $\text{int}(\partial K)$ to $\mathbb{S}^{n-1}_- := \mathbb{S}^{n-1} \cap \mathbb{R}^n_{< 0}$.*

We now specialize some properties of the support function to $C^2_+(\mathcal{K}^n_*)$.

**Lemma 4.8.** *If $K \in C^2_+(\mathcal{K}^n_*)$, then $\text{dom}(\sigma_K) = \text{int}(\mathbb{R}^n_{\leq 0}) \cup \{0\}$.*

*Proof.* Take $u \neq 0$ in $\text{dom}(\sigma_K)$, then it must be an outward normal vector to $\text{int}(\partial K)$, hence it is in $\mathbb{S}^{n-1}_-$. Then $\text{dom}(\sigma_K) \subset \text{int}(\mathbb{R}^n_{\leq 0}) \cup \{0\}$. Now, for $u \in \mathbb{R}^n_{< 0}$, normalize it to make it unitary by letting $v = u/|u|$, then $v \in \mathbb{S}^{n-1}_-$ and thus it must be a normal vector form some $x \in \text{int}(\partial K)$, hence the support function evaluated at $v$ is finite, and in consequence $\sigma_K(u)$ is finite too. $\qquad\square$

**Remark 4.9.** *Following Schneider (2014, Section 2.5) the condition $K \in C^2_+(\mathcal{K}^n_*)$ is equivalent to assuming the principal curvatures of $\partial K$ to be non-zero. It also follows that*

$$\sigma_K(u)|_{\mathbb{S}^{n-1}_-} = \langle \mathbf{u}_K^{-1}(u), u \rangle,$$

*and moreover, $\sigma_K$ is of class $C^2$.*

**Remark 4.10.** *Let $\ell \in \mathcal{L}_n$ be a proper loss function. By definition we see that Remark 4.9 implies* $\mathrm{spr}(\ell) \in C^2_+(\mathcal{K}^n_*)$ *(since $M_\ell = \partial(\mathrm{spr}(\ell))$).*

**Definition 4.11.** *Let $K, L \in C^2_+(\mathcal{K}^n_*)$. We say that $L$ slides freely inside $K$ if to each boundary point $x$ of $K$, there exists a translation vector $t \in \mathbb{R}^n$, such that $x \in L + t \subset K$.*

**Theorem 4.12.** *Let $K, L \in C^2_+(\mathcal{K}^n_*)$. $L$ is a summand of $K$, then $L$ slides freely inside $K$.*

*Proof.* Suppose that there exists $M \in C^2_+(\mathcal{K}^n_*)$ such that $K = L + M$. Let $x \in \partial K$. Then there are $l \in L$ and $m \in M$ such that

$$x = l + m.$$

Thus, $x \in L + m \subset L + M = K$. $\qquad\square$

**Remark 4.13.** *For a general convex set $L$, if $L$ is a summand of $K \in C^2_+(\mathcal{K}^n_*)$ we see that the previous proof holds an we conclude that $L$ slides freely inside $K$; note however that this imposes restrictions on possible sets $L$. One of this consequences is that the principal curvatures of $\partial L$ must be positive as can be seen from a second fundamental form comparison and Theorem 3.8.*

**Lemma 4.14.** *Let $K, L \in C^2_+(\mathcal{K}^n_*)$ and suppose that $f(\cdot) = \sigma_K(\cdot) - \sigma_L(\cdot)$ is convex. Then the set*

$$M = \{x \in \mathbb{R}^n \mid \langle x, u \rangle \le f(u) \text{ for all } u \in \mathbb{R}^n\},$$

*is in $C^2_+(\mathcal{K}^n_*)$, and it is such that $K = M + L$, that is, $L$ and $M$ are summands of $K$.*

*Proof.* From Lemma 4.8, the domain of $f$ is $\mathbb{R}^n_{<0} \cup \{0\}$, i.e., $f \colon \mathbb{R}^n_{<0} \cup \{0\} \longrightarrow \mathbb{R}$ is convex. Thus it is the support function of $M$ (by Lemma 4.4). That is, $f(\cdot) = \sigma_M(\cdot)$.

Therefore we have $\sigma_M = \sigma_K - \sigma_L$, and hence $K = M + L$. Note that $M$ is a summand of $K$, then using Theorem 4.12 we know that $M$ slides freely inside $K$, and since $\partial K$ has positive principal curvatures then $\partial M$ does too (Remark 4.13). Since $\sigma_M$ is of class $C^2$, then $M$ has to be in $C^2_+(\mathcal{K}^n_*)$. $\qquad\square$

**Theorem 4.15.** *[(Schneider, 2014, Theorem 1.5.2)] Let $D \subset \mathbb{R}^n$ convex and let $f \colon D \longrightarrow \mathbb{R}$ be a continuous function. Suppose that for each point $x_0 \in D$ there are an affine function $g$ on $\mathbb{R}^n$ and a neighborhood $U$ of $x_0$ such that $f(x_0) = g(x_0)$ and $f \ge g$ in $U \cap D$. Then $f$ is convex.*

**Definition 4.16.** *We say that $L$ is locally embeddable in $K$ if for all $x \in \partial K$, there is a $y \in L$ and a neighborhood $U$ of $y$, such that $(L \cap U) + x - y \subset K$.*

**Theorem 4.17.** *Let $K, L \in C^2_+(\mathcal{K}^n_*)$ and $L$ strictly convex. If $L$ is locally embeddable in $K$, then $L$ is a summand of $K$.*

*Proof.* Let $u_0 \in \mathbb{S}^{n-1}_-$ and $x_0 \in \partial K$ be a point such that $\mathbf{u}(x_0) = u_0$. Since $L$ is locally embeddable in $K$ there are $y_0 \in L$ and a neighborhood $U_0$ of $y_0$ such that $(L \cap U_0) + x_0 - y_0 \subset K$. Since $\mathbf{u}^{-1}_L \colon \mathbb{S}^{n-1}_- \longrightarrow \partial L$ is continuous, there exists a neighborhood $V_0$ of $u_0$ such that $\mathbf{u}^{-1}_K(V_0) \subset U_0$. Then it follows that $\sigma(L + x_0 - y_0, u_0) = \sigma(K, u_0)$ and $\sigma(L + x_0 - y_0, u) \le \sigma(K, u)$ for all $u \in V_0$ by Lemma 4.2.

Let $f(\cdot) = \sigma(K, \cdot) - \sigma(L, \cdot)$ (this is defined on $\mathbb{R}^n_{\le 0} \cup \{0\}$ and is positively homogeneous), and $g(\cdot) = \langle x - y, \cdot \rangle$. Then, clearly, we have

(i) $f(u_0) = g(u_0)$, since

$$\begin{aligned}
f(u_0) &= \sigma(K, u_0) - \sigma(L, u_0) \\
&= \sigma(K, u_0) - \sigma(L + x_0 - y_0, u_0) + \langle x_0 - y_0, u_0 \rangle \\
&= g(u_0).
\end{aligned}$$

(ii) $f \ge g$ on $V_0$,

$$\begin{aligned}
f(u) &= \sigma(K, u) - \sigma(L, u) \\
&= \sigma(K, u) - \sigma(L + x_0 - y_0, u) + \langle x_0 - y_0, u \rangle \\
&\ge g(u).
\end{aligned}$$

It follows by Theorem 4.15 that $f$ is convex, and by Lemma 4.14 we conclude that $L$ is a summand of $K$. $\square$

The following lemma is a direct consequence of the characterization of mixability in Theorem 3.8 and Definition 4.16.

**Lemma 4.18.** *Let $\ell \in \mathcal{L}_n$ be a proper loss. For $\eta > 0$, if $\ell$ is $\eta$-mixable then $\mathrm{spr}(\eta\ell)$ is locally embeddable in $\mathrm{spr}(\lambda)$.*

**Lemma 4.19.** *If $\ell$ is $\eta$-mixable then $\mathrm{spr}(\eta\ell)$ slides freely inside $\mathrm{spr}(\lambda)$.*

*Proof.* Let $\ell$ be $\eta$-mixable, then Lemma 4.18 implies $\mathrm{spr}(\eta\ell)$ it is locally embeddable in $\mathrm{spr}(\lambda)$. Then Theorem 4.17 implies it is a summand and Theorem 4.12 implies it slides freely inside $\mathrm{spr}(\lambda)$. $\square$

**Corollary 4.20.** *Let $\ell$ be a $\eta$-mixable proper loss. Then $\mathrm{spr}(\eta\ell) \in C_+^2(\mathcal{K}_*^n)$ and it slides freely inside $\mathrm{spr}(\lambda)$ ($\lambda$ is the log loss). Additionally, there exists $M \in C_+^2(\mathcal{K}_*^n)$ such that*

$$\mathrm{spr}(\lambda) = \mathrm{spr}(\eta\ell) + M.$$

*Moreover, $\partial M$ can be regarded as $\varrho(\Delta^n)$ for a 1-mixable proper loss $\varrho$.*

*Proof.* Since $\ell$ is an $\eta$-mixable proper loss function, $\eta\ell$ is also a proper loss function and hence $\mathrm{spr}(\eta\ell) \in C_+^2(\mathcal{K}_*^n)$ (Remark 4.10). Theorem 3.8 implies that $\mathrm{spr}(\eta\ell)$ is locally embeddable in $\mathrm{spr}(\lambda)$. From Theorem 4.17 we know that $\mathrm{spr}(\eta\ell)$ is a summand of $\mathrm{spr}(\lambda)$, which proves the existence of $M$. As a consequence, $M$ is a convex set with recession cone $\mathbb{R}_{\geq 0}^n$ (Lemma 4.14). By applying (Williamson & Cranko, 2022, Proposition 21) we can regard $\partial M$ as the image of a proper loss function $\varrho$, which since $\mathrm{spr}(\varrho)$ is a summand of $\mathrm{spr}(\lambda)$ it is 1-mixable (Lemma 4.14). $\square$

We now state (Schneider, 2014, Theorem 2.5.4) adapted to our setting which will be helpful to relate our work to (van Erven et al., 2012).

**Theorem 4.21.** *Let $K, L \in C_+^2(\mathcal{K}_*^n)$. Let $h^M(x)$ denote the second fundamental form of $M$ at $x$ with respect to $\mathbf{u}$ (see (A.2)). The following are equivalent:*

*(i) $h^{\partial L}(x) \geq h^{\partial K}(y)$ for all pairs of points $x$ and $y$ at which $\mathbf{u}(x) = \mathbf{u}(y)$.*

*(ii) $\sigma_K - \sigma_L$ is a support function.*

Since $\Delta^n$ is an affine manifold, the geodesics in $\Delta^n$ are simply straight lines. This allows to define convexity of functions defined on $\Delta^n$ in the usual way we do for functions on $\mathbb{R}^n$. The following theorem connects and reconciles our results to those in (van Erven et al., 2012). More precisely, we create a bridge between our results and (van Erven et al., 2012, Theorem 10).

**Theorem 4.22.** *Let $\ell \in \mathcal{L}_n$ be proper loss. Let $\eta > 0$, then $\ell$ is $\eta$-mixable if and only if $\eta\underline{L}^\ell(\cdot) - \underline{L}^\lambda(\cdot)$ is convex on $\mathrm{int}(\Delta^n)$, where $\underline{L}^\varrho(\cdot)$ denotes the Bayes risk of the loss function $\varrho$ (Definition 1.3) and $\lambda$ denotes the log loss.*

*Proof.* Suppose that $\ell$ is a proper loss in $\mathcal{L}_n$ which is $\eta$-mixable. By Lemma 4.19 $\mathrm{spr}(\eta\ell)$ slides freely inside $\mathrm{spr}(\lambda)$ and in particular $h^{\eta\ell}(\ell(p)) \geq h^\lambda(\lambda(p))$. By Theorem 4.21 it follows that $\sigma_{\mathrm{spr}(\lambda)} - \sigma_{\mathrm{spr}(\eta\ell)}$ is a support function with domain $\mathbb{R}_{<0}^n \cup \{0\}$, in particular it is convex on its interior. Let $u \in \mathbb{R}_{<0}^n$, such that the outward normal vector of $\ell(\Delta^n)$ and $\lambda(\Delta^n)$ at $\ell(p)$ and $\lambda(p)$, respectively, is $u$. Then we have for $x = -p \in \Delta^n$,

$$\begin{aligned}
\sigma_{\mathrm{spr}(\lambda)}(x) - \sigma_{\mathrm{spr}(\eta\ell)}(x) &= |x|(\sigma_{\mathrm{spr}(\lambda)}(x/|x|) - \sigma_{\mathrm{spr}(\eta\ell)}(x/|x|)) \\
&= |x|((\langle\lambda(p), x/|x|\rangle - \langle\eta\ell(p), x/|x|\rangle) \\
&= |p|((\langle\lambda(p), -p/|p|\rangle - \langle\eta\ell(p), -p/|p|\rangle) \\
&= \langle\lambda(p), -p\rangle - \langle\eta\ell(p), -p\rangle \\
&= -\langle\lambda(p), p\rangle + \langle\eta\ell(p), p\rangle \\
&= -\underline{L}^\lambda(p) + \eta\underline{L}^\ell(p),
\end{aligned}$$

which proves the claim. $\square$

Suppose now that for given $\ell \in \mathcal{L}_n$ proper, there exists a $\eta > 0$ such that $\mathrm{spr}(\eta\ell)$ slides freely inside $\mathrm{spr}(\lambda)$. Note that in particular this implies that $\mathrm{spr}(\eta\ell)$ is locally embeddable in $\mathrm{spr}(\lambda)$, and hence for each $p \in \mathrm{int}(\Delta^n)$ we have

$$h^{\eta\ell}(\eta\ell(p)) - h^\lambda(\lambda(p)) \geq 0,$$

which by (3.10) and Lemma 3.6 implies that $\ell$ is $\eta$-mixable. Thus combining this with Lemma 4.19 we obtain the following characterization of mixability of proper (sufficiently differentiable) loss functions.

**Theorem 4.23.** *Let $\ell \in \mathcal{L}_n$ be proper. $\ell$ is $\eta$-mixable if and only if $\mathrm{spr}(\eta\ell)$ slides freely inside $\mathrm{spr}(\lambda)$, where $\lambda$ denotes the log loss.*

In general, the set $\mathcal{L}$ provides a family of loss functions with appealing properties. Arguably, one of the most important properties is that given $\ell \in \mathcal{L}$, if we assume that $\ell$ is proper then we know its principal curvatures are strictly positive. This is a strong and useful geometric feature. For example, in (Williamson & Cranko, 2022) the notion of a "inverse loss" called the *anti-polar loss* was investigated. Given $\ell$ a proper loss (in the sense of (Williamson & Cranko, 2022), which are not necessarily smooth), they consider the 0-homogeneous extension of $\ell$ (see Remark 26 in (Williamson & Cranko, 2022)), defined on $\mathbb{R}_{>0}^n$ and given by

$$\ell^{\mathrm{ext}}(p) := \ell\left(\frac{p}{\|p\|_1}\right),$$

where $\|p\|_1 = p_1 + \ldots + p_n$. For the following we simply denote $\ell^{\mathrm{ext}}$ by $\ell$. In (Williamson & Cranko, 2022, Proposition 29) it is shown that there exists a map $\ell^\diamond \colon \mathbb{R}_{>0} \longrightarrow \mathbb{R}_{\geq 0}^n$ such that

$$\ell(p) = (\ell \circ \ell^\diamond \circ \ell)(p)$$
$$\ell^\diamond(x) = (\ell^\diamond \circ \ell \circ \ell^\diamond)(x),$$

for all $x, p \in \mathbb{R}_{>0}^n$. The map $\ell^\diamond$ is called the *anti-polar* loss of $\ell$. For the family of admissible loss function $\mathcal{L}$ considered in this work, we exploit the differentiability conditions to obtain in a straightforward way an inverse loss defined on $\ell(\mathrm{int}(\Delta^n))$. To see this, suppose that $\ell \in \mathcal{L}$ is proper. Since this is equivalent to saying that $\mathrm{spr}(\ell)$ is in $C_+^2(\mathcal{K}_*^n)$, meaning that the map $\mathbf{u}_{\mathrm{spr}(\ell)}$ is $C^1$ diffeomorphism. Then we can define the map $\ell^{-1} \colon \ell(\mathrm{int}(\Delta^n)) \longrightarrow \mathrm{int}(\Delta^n)$ by

$$\ell^{-1}(x) := \frac{\mathbf{u}_{\partial\mathrm{spr}(\ell)}(x)}{\|\mathbf{u}_{\mathrm{spr}(\ell)}(x)\|_1},$$

which is the inverse of the map $\ell \colon \mathrm{int}(\Delta^n) \longrightarrow \ell(\mathrm{int}(\Delta^n))$. Recall that $\mathbf{u}_{\partial\mathrm{spr}(\ell)}(x)$ is nothing else than the unit normal vector (pointing towards $\mathbb{R}_{\geq 0}^n$) at $x \in \ell(\mathrm{int}(\Delta^n))$.

It is of interest of finding parametrizations (or links) that simplify the expression of a given proper loss $\ell$. At a theoretical level there are potentially many ways to to this. Notably we have at hand the notion of *canonical link* in (Williamson et al., 2016) (or see Section 2.7 above for $n = 2$). As an example of other ways to obtain nice links we have Lemma 3.3 above, which gives a nice expression in coordinates (as the form of a graph) of $\ell$. Unfortunately, to obtain that results one makes uses of the inverse function theorem which does not provide an explicit inverse but rather its existence.

# 5    Conclusions

We summarize the main messages of this work.

- Since mixable loss functions are of great importance in prediction games, it is desirable to understand them from different perspectives. Inspired by the work of Vovk (2015), in Section 2 we studied binary loss functions from the point of view of differential geometry, hence restricting to loss functions in $\mathcal{L}$ (Definition 2.1). To do this, we re-interpret properness as a geometric property, namely, a loss function $\ell \in \mathcal{L}$ is proper if and only if

  - the normal vector (belonging to $\mathbb{R}_{\geq 0}^2$) to $M_\ell = \ell(\mathrm{int}(\Delta^2))$ at $\ell(p)$ is $\frac{p}{|p|}$, for any $p \in \mathrm{int}(\Delta^2)$, and

   – the loss curve $\ell(\mathrm{int}(\Delta^2))$ has positive curvature (with respect to $\frac{p}{|p|}$).

Having this framework at hand, we characterized mixability and fundamentality of a proper loss $\ell \in \mathcal{L}$, as a curvature comparison to the log loss $\ell_{\log}$ (cf. (Vovk, 2015)).

- In Section 3, we extended the geometric characterization of proper loss functions to higher dimensions, and obtained the corresponding interpretation of mixability as a geometric comparison (now in terms of the principal curvatures of the "loss surface"). This comparison is done by using the second fundamental forms of the "loss surfaces".

- The main goal of Section 4 is to re-interpret the geometric results in Section 3 from the point of view of convex geometry. The main result in this part is a new characterization of $\eta$-mixability of a proper loss function $\ell \in \mathcal{L}$, as $\mathrm{spr}(\eta\ell)$ sliding freely inside $\mathrm{spr}(\ell_{\log})$ (in general dimension). This provides an intuitive and geometric way to interpret mixability.

- Since the results obtained in this work are in terms of curvature, it was necessary to re-interpret well known properties of loss functions in the language of differential geometry. Although slightly tedious, this allowed the reconciliation of the results obtained by Vovk (2015) for $n = 2$ and by van Erven et al. (2012) for $n \geq 2$.

- Theorem 4.22 connects our results to (van Erven et al., 2012, Theorem 10) as follows. In (van Erven et al., 2012, Theorem 10) the following statements are proven to be equivalent:

   (i) a proper loss $\ell \in L$ is $\eta$-mixable,

   (ii) $\eta H\underline{\widetilde{L}}(t) - H\underline{\widetilde{L}}_{\log}(t)$ is positive semi-definite for all $t \in \Phi_{\mathrm{std}}^{-1}(\mathrm{int}(\Delta^n))$, where $HF(t)$ denotes the Hessian of $F$ at $t$,

   (iii) $\eta\underline{L}(p) - \underline{L}_{\log}(p)$ is convex on $\mathrm{int}(\Delta^n)$, and

   (iv) $\eta\underline{\widetilde{L}}(p) - \underline{\widetilde{L}}_{\log}(p)$ is convex on $\Phi_{\mathrm{std}}^{-1}(\mathrm{int}(\Delta^n))$.

There, they first proved the equivalence of (i) and (ii), which is the result of a long direct computation done very carefully. The equivalence between (iii) and (iv) is straightforward. To connect these two sets of equivalences, standard convex geometry is used to prove the equivalence of (ii) and (iii). Note that the statements (ii) and (iv) make reference to a precise choice of parametrization of $\Delta^n$ (i.e., the standard parametrization $\Phi_{\mathrm{std}}$), therefore, the work presented here is naturally not related to these statements but rather to (i) and (iii), whose equivalence can be considered to be the content of Sections 3 and 4. Determining whether this new approach provides a simplification of the computations in (van Erven et al., 2012) or not, strongly depends on the differential geometry and convex geometry background of the reader. This work should be considered as complementing the understanding of mixable loss functions and providing a new geometric insight into them.

## Acknowledgements

This work was supported by the Deutsche Forschungsgemeinschaft under Germany's Excellence Strategy — EXC number 2064/1 — Project number 390727645.

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

# A  Differential Geometry

In this part we provide a brief summary of the concepts of differential geometry that are used in this work (we assume the reader has some familiarity with the topic although we try to put emphasis on the intuition). We do not intend to give a comprehensive introduction to the topic. Most of the material can be found in almost any differential geometry book, however, we recommend (and when possible use the notation of) (do Carmo, 2016) and (Lee, 2018).

## A.1  Curvature of Curves

A *parametrized curve* is a differentiable map $\alpha\colon (a,b) \to \mathbb{R}^n$, $(a < b)$. We are interested in studying the *geometry* of parametrized curves. For this it would be useful to restrict our discussions to curves with a well defined tangent line at every point $\alpha(t)$ for $t \in (a,b)$ (i.e., with non-vanishing $\alpha'(t)$). These curves are called *regular*. Let $\varphi\colon (a,b) \to (c,d)$ be a diffeomorphism, the curve $\beta = \alpha(\varphi(s))$ is a *reparametrization* of $\alpha$. Note that in this case $\alpha((a,b)) = \beta((c,d))$. The image $M = \alpha((a,b))$ is a 1-dimensional differentiable manifold in $\mathbb{R}^n$ (for this it is essential to restrict to regular curves). The study of curves is of particular importance since some aspects are carried to the study of the geometry of general hypersurfaces in $\mathbb{R}^n$.

Typically, curvature is defined for curves *parametrized by arc-length* meaning that $|\beta'(s)| = 1$ for all $s \in (c,d)$ (and a regular curve can always be parametrized this way). For these types of curves, the *curvature of $\beta$ at* $\beta(s)$ is defined as the length of $\beta''(s)$, which measures "how much" a curve "curves". However, this notion does not give information about the direction on which a curve is "curving". We start looking at the case $n = 2$. We define the *signed curvature* of a general curve $\alpha(t) = (x_1(t), x_2(t))$ by (cf. (1.5))

$$\kappa_\alpha(t) := \frac{x_1'(t)x_2''(t) - x_1''(t)x_2(t)}{\left(x_1'(t)^2 + x_2'(t)^2\right)^{3/2}}.$$

It can be checked that $|\kappa(t)|$ coincides with the curvature of $\alpha$ when parametrized by arc-length (at the corresponding point), the signed curvature is well defined up to a sign (the sign will change if we consider a reparametrization that reverses the order of $(a,b)$, for example a curve defined on $(-b,-a)$ given by $\beta(s) = \alpha(-s)$), which motivates the discussion in Section 1.5.

For example, suppose that a planar curve is defined by a function $f\colon (a,b) \longrightarrow \mathbb{R}$ is the following way:

$$\alpha(t) = (t, f(t)),$$

for $t \in (a,b)$. A quick computation gives

$$\kappa_\alpha(t) = \frac{f''(t)}{(1 + f'(t)^2)^{3/2}}. \tag{A.1}$$

Given a regular curve $\alpha\colon (a,b) \longrightarrow \mathbb{R}^3$ as above and a real number $\eta \neq 0$, it is straightforward to see that the curve $\beta(t) = \eta\alpha(t)$ is also a regular curve and its signed curvature is given by

$$\kappa_\beta(t) = \frac{\eta^2 x_1'(t)x_2''(t) - \eta^2 x_1''(t)x_2(t)}{(\eta^2 x_1'(t)^2 + \eta^2 x_2'(t)^2)^{3/2}} = \frac{1}{\eta}\kappa_\alpha(t).$$

The notion of signed curvature can be extended to curves in manifolds sitting inside $\mathbb{R}^n$ (see for example (Lee, 2018, Chapter 8)). For $\alpha(-\varepsilon, \varepsilon) \longrightarrow \mathbb{R}^n$ parametrized by arc-length, the *signed curvature* (with respect to $\mathbf{n}$) $\kappa_\alpha^+$ of $\alpha$ at $p = \alpha(0)$ is given by $\kappa_\alpha^+(0) = \langle \mathbf{n}, \alpha''(0) \rangle$. It can be shown that this definition agrees with the one we gave for $n = 2$.

### A.2 Geometry of hypersurfaces in $\mathbb{R}^n$

Let $M$ be a differentiable hypersurface inside $\mathbb{R}^n$ of class $C^k$ (i.e., a $n-1$-dimensional $C^k$ manifold). By this we mean that for each $p \in M$ there is an open set $U \subset \mathbb{R}^{n-1}$ and a $C^k$ injective map $\Phi \colon U \longrightarrow M$ (called a *parametrization of $M$ around $p$*). For each $x \in U$, $\{\partial_1 \Phi(x), ..., \partial_{n-1} \Phi(x)\}$ forms a basis for the tangent space $T_q M$ ($q = \Phi(x)$) to $M$ at $q$. Since $\Phi(U) \subset \mathbb{R}^n$ we can consider the induced metric on $M$ by the Euclidean metric in $\mathbb{R}^n$ (denoted by $\langle \cdot, \cdot \rangle$). This is a Riemannian metric on $M$ given on the coordinates given by $\Phi$ by the matrix

$$g_{ij}(x) = \langle \partial_i \Phi(x), \partial_j \Phi(x) \rangle,$$

for $x \in U$. The metric $g$ allows us to define the length of a curves in $M$.

In general, if a manifold $M$ of dimension $n-1$ is sitting inside an $n$-dimensional Riemannian manifold $\overline{M}$ (and $M$ is endowed with the induced metric from $M$) the second fundamental form carries the information on how $M$ is "curved" inside $\overline{M}$. Let $\bar{g}$ be the metric on $\overline{M}$ and $g$ the induced metric on $M$ by $\bar{g}$. Let $\overline{\nabla}$ denote the Levi–Civita connection of $\bar{g}$. Let $\mathbf{n}$ be a smooth unit normal vector field to $M$ (that is $\mathbf{n}(p)$ is perpendicular to $T_p M$ for each $p \in M$). The *scalar second fundamental form of $M$ with respect to $\mathbf{n}$* is the covariant 2-tensor $h$ on $M$ defined as

$$h(X, Y) = \langle \mathbf{n}, \overline{\nabla}_X Y \rangle = -\langle \overline{\nabla}_X \mathbf{n}, Y \rangle. \tag{A.2}$$

for $X, Y$ tangent vectors to $M$. Note that for a hypersurface, at each point we have exactly to unit normal vectors to $M$ at $p$, thus the scalar second fundamental form is well-defined up to a sign. Fixing a point $p \in M$ and an orthonormal basis $\{E_1, ..., E_{n-1}\}$ for the tangent space at $p$ $T_p M$, the eigenvalues of the matrix given by $h_{ij} = h(E_i, E_j)$ for $i, j = 1, ..., n-1$ are called the *principal curvatures* of $M$ at $p$ and the corresponding eigenspaces are called the *principal directions*. For details of the above see Chapter 8 in (Lee, 2018).

When $\overline{M} = \mathbb{R}^n$ and $M$ is parametrized by $\Phi \colon U \subset \mathbb{R}^{n-1} \longrightarrow M \subset \mathbb{R}^n$, with respect to the local frame $\{\partial_1 \Phi, ..., \partial_{n-1} \Phi\}$ of $\Phi(U)$, the scalar second fundamental form with respect to a normal unit vector field $\mathbf{n}$ is given by ((Lee, 2018, Proposition 8.23))

$$h_{ij} = h(\partial_i \Phi, \partial_j \Phi) = \langle \partial_{ij} \Phi, \mathbf{n} \rangle, \tag{A.3}$$

for $i, j = 1, ..., n-1$.

Given any $p \in M$ and $v \in T_p M$, there a geodesic $\gamma_V \colon (a, b) \longrightarrow M$ of $M$ passing through $p$ with velocity $v$ at $p$. Let $M_1$ and $M_2$ be two hypersurfaces in $\mathbb{R}^{n+1}$ tangent at a point $p \in M_1 \cap M_2$. Choose a normal vector $\mathbf{n}$ and suppose that $M_1$ lies above $M_2$ (with respect to $\mathbf{n}$). We have the following lemma from (Lee, 2018).

With the previous lemma we can obtain a comparison result for manifolds with positive principal curvatures.

**Lemma A.1.** *Suppose that $M_1$ and $M_2$ are tangent at $p \in M_1 \cap M_2$ and fix a normal vector $\mathbf{n}$ at $p$. Suppose that $M_1$ and $M_2$ have positive principal curvatures at $p$. Then $h_1(v, v) \geq h_2(v, v)$ for all $v \in T_p M$ if and only if $M_1$ lies above $M_2$ (with respect to $\mathbf{n}$) locally around $p$.*

*Proof.* First we make the following observation. Suppose that $M$ is a smooth hypersurface in $\mathbb{R}^n$ and we have a regular curve $\alpha \colon (-\varepsilon, \varepsilon) \longrightarrow M$ such that $\alpha(0) = p$ and $\alpha'(0) = v$ for some $p \in M$ and $v \in T_p M$. Then, letting $h$ denote the second fundamental form of $M$ from (A.2) we have

$$
\begin{aligned}
h(v, v) &= -\langle \overline{\nabla}_v n, v \rangle \\
&= -\langle \frac{d(\mathbf{n} \circ \alpha)}{dt}(t), \alpha'(t) \rangle \Big|_{t=0} \\
&= \langle (\mathbf{n} \circ \alpha)(t), \alpha''(t) \rangle \Big|_{t=0} \\
&= \langle \mathbf{n}, \alpha''(0) \rangle.
\end{aligned}
$$

Thus, if $\alpha$ is parametrized by arc-length, $h(v, v) = \langle \mathbf{n}, \alpha''(0) \rangle = \kappa_\alpha^+(0)$.

Suppose $M_1$ lies above $M_2$ are tangent at $p$ and let $v \in T_p M_1 = T_p M_2$ with $|v| = 1$. Then we can intersect $M_1$ and $M_2$ with the plane generated by $v$ and $\mathbf{n}$. Then we obtain two curves $\alpha_1$ and $\alpha_2$ on $M_1$ and $M_2$, respectively, such that $\alpha_i(0) = p$ and $\alpha'(0) = v$ for $i = 1, 2$. Moreover, we can assume that these curves are parametrized by arc-length so its Euclidean curvature is given by $\langle \alpha_i''(0), \mathbf{n} \rangle$. Since we can regard these curves as planar curves, there are functions $f_1$ and $f_2$ such that the curves $\alpha_1$ and $\alpha_2$ are represented in the plane $\langle v, \mathbf{n} \rangle$ by the curves

$$\gamma_1(x) = (x, f_1(x))$$
$$\gamma_2(x) = (x, f_2(x)),$$

with $f_i = (0)$, $f_i'(0) = v$, $f_i''(0) > 0$ (since $M_1$ and $M_2$ have positive principal curvatures at $p$) for $i = 1, 2$. By construction $\kappa_{\gamma_1}^+(0) = f_i''(0)$ and by definition $\kappa_{\gamma_i}^+(0) = \langle \alpha_i''(0), \mathbf{n} \rangle$, for $i = 1, 2$.

If $M_1$ lies above $M_2$ at $p$, then $f_1''(0) > f_2''(0)$ and hence $\kappa_{\gamma_1}^+(0) \geq \kappa_{\gamma_2}^+(0)$, which is equivalent to $h_1(v, v) \geq h_2(v, v)$ for any $v \in T_p M$ with $|v| = 1$. Let $w \neq 0 \in T_p M$ be arbitrary, then

$$h_1(w, w) = |w|^2 h_1\left(\frac{w}{|w|}, \frac{w}{|w|}\right) \geq |w|^2 h_2\left(\frac{w}{|w|}, \frac{w}{|w|}\right) = h_2(w, w), \tag{A.4}$$

as claimed.

Conversely if (A.4) holds, then we see that in particular holds for unitary $v$, which ultimately means that $f_1''(0) \geq f_2''(0)$ for all unitary $v \in T_p M$. This implies that $M_1$ lies above $M_2$. $\qquad\square$

We present the following instructive example.

**Example A.2.** *Consider the differentiable function $f_\kappa(x, y) = \kappa(x^2 + y^2)$ with $\kappa > 0$, and let $M_\kappa = \{(x, y, f_\kappa(x, y)) \mid (x, y) \in B_1(0)\}$. We choose the parametrization $\Phi_\kappa(x) = (x, f_\kappa(x))$ of $M_\kappa$ and compute the scalar second fundamental form of $M_\kappa$ at $p = (0, 0, 0)$ in these coordinates. We have*

$$\partial_x \Phi(x, y) = (1, 0, 2\kappa x),$$
$$\partial_y \Phi(x, y) = (0, 1, 2\kappa y),$$
$$\partial_{xx} \Phi(x, y) = (0, 0, 2\kappa),$$
$$\partial_{xy} \Phi(x, y) = (0, 0, 0),$$
$$\partial_{yy} \Phi(x, y) = (0, 0, 2),$$

*thus from (A.3) at the point $\Phi_\kappa(0, 0) = (0, 0, 0)$, the scalar second fundamental form of $M_\kappa$ with respect to $\mathbf{n} = (0, 0, 1)$ is given by*

$$[h_{f_\kappa}](p) = \begin{pmatrix} 2\kappa & 0 \\ 0 & 2\kappa \end{pmatrix},$$

*and in particular for $\kappa = 1$ we have*

$$[h_{f_1}](p) = \begin{pmatrix} 2 & 0 \\ 0 & 2 \end{pmatrix}.$$

*Thus, clearly we have*

$$[h_{f_\kappa}](0) - [h_{f_1}](0) = \begin{pmatrix} 2\kappa - 2 & 0 \\ 0 & 2\kappa - 2 \end{pmatrix} \tag{A.5}$$

*which is positive definite if and only if $\kappa > 1$ (when $M_\kappa$ lies inside $M_1$ and are tangent at $p$).*

**Remark A.3.** *We stress a technical observation. The comparison (A.5) in Example A.2 is valid since regardless of the value of $\kappa$, $\partial_x \Phi_\kappa(0, 0)$ and $\partial_y \Phi_\kappa(0, 0)$ are the same, meaning that we can identify the tangent spaces to $M_\kappa$ and $M_1$ at $p$ for all $\kappa$, and the basis for them is given by $\{\partial_x \Phi_1(0, 0), \partial_y \Phi_\kappa(0, 0)\}$. In general this is not necessarily the case so one should perform a change of basis before comparing the second fundamental forms.*

