# OpenReview forum: "The Geometry of Mixability"
_TMLR — Accepted by TMLR_

### Review · Reviewer_ABr5 · 2023-04-30

**Summary Of Contributions:**

The authors of this paper provide new characterizations about the mixability and fundamentality of proper loss functions in the setting of prediction with expert advice, from a geometric perspective. In particular, authors reveal the fact that both properties of the loss function are geometric and should not depend on the ambient coordinate system.

**Audience:**

Yes

**Broader Impact Concerns:**

I have no such concerns.

**Claims And Evidence:**

No

**Requested Changes:**

Critical:
1. More explanations about the advantage of the proposed geometric interpretation are needed.
2. Organization of contents should be adjusted. Lemmas and proofs that are not important to the understanding of the main results should be deferred to the appendix.
3. Technical errors need to be fixed:
- In the definition of standard parametrization of $\Delta^n$, its domain shouldn't be $\Delta^{n-1}$, otherwise it always maps $(t_1,...,t_{n-1})$ to $(t_1,...t_{n-1},0)$. As this notion is used throughout this paper, such an error may affect the proofs.
- In Definition 2.1(ii), there is a  $n_{\ell(p)}$ that is never mentioned or defined before, which makes the definition of map $\textbf{n}$ ambiguous.
- In Definition 2.1(iii), there is a $\textbf{n}(p)$, where $p$ as the argument should be in $\ell(int(\Delta^2))$ as in (ii), but in fact it's in $int(\Delta^2)$. This is confusing.
- In chapter 4, there is no definition of $C^2_{-}(\mathcal K^n_{*})$ which first appears in Lemma 4.14
4. Need to add more contexts or intuitive interpretations regarding:
- The setting of prediction with expert advice, including what kind of predictions are made by the learner and experts, what kind of outcomes are revealed by nature, and how the loss is incurred.
- The formal definition of regret and $\eta-$mixable games. A clearer statement about the $O(\log n/\eta)$ regret bound in such games is also needed.
- Certain properties of loss functions, including properness, mixability and fundamentality.

Not critical:
1. Typos and bad notations. Here are a few of them:
- $n$ is used as both the number of experts and the number of possible outcomes. I think these two values are different and should be denoted differently.
- In the definition of probability simplex (on page 3 and 7), its points should have non-negative coordinates.
- In the first formula on page 4, there should be indicator functions.
- In the last sentence of the first paragraph of section 1.5, "the signed curvature of $\kappa$" should be "the signed curvature of $\alpha$".
- In the first line of chapter 3, "$\mathbb R^n \geq 0$" should be "$\mathbb R^n_{\geq 0}$".
- In Definition 4.11, "$C^2_{+}(\mathcal K^n)$" should probably be "$C^2_{+}(\mathcal K^n_{*})$".

**Strengths And Weaknesses:**

Strengths:
1. Authors provide plenty of comparisons and connections to previous works.

Weaknesses:
1. It's not so apparent to me the benefit brought by characterizing the mixability of a loss function by the ability of its superprediction set to slide freely inside that of the log loss. Indeed such characterization gets rid of the obscure notion of exponential projection, but that doesn't seem enough, say it still involves the notion of superprediction set, which remains mysterious throughout. So more explanations about the strengths of this new characterization are needed.
2. There is a lack of context and intuitive interpretations of relevant technical terms. For example, the motivation for loss being proper and fundamental is not present. More details can be found in the next section.
3. There are some technical errors that appear to have a negative effect on general reading and understanding. For example, the definition of standard parameterization is incorrect throughout the paper. In particular, I suspect that the current inapplicable definition of standard parametrization might affect the correctness of some technical results in this paper. More details can be found in the next section.
4. The organization of the contents in this paper is hard to follow. For example, there are too many technical lemmas with lengthy proofs in the main text, which makes it hard to capture the main results and follow the line of their derivation.

---

### Review · Reviewer_yBnk · 2023-05-30

**Summary Of Contributions:**

In online learning/sequential prediction with a finite number of experts, it is well known that constant regret (in horizon) is achievable for certain classes of losses, e.g., the special case of exp-concave losses. A general characterization of such class via mixable losses was studied by Vovk in the context of his aggregation algorithm. However, unlike exp-concavity which is geometric, the original definition of mixable losses was a bit cumbersome — it’s a for-all guarantee where one must point-wise “guess” an aggregation/“mean” function. Various simpler-but-equivalent characterizations have been sought and proposed, notably a clean formula from Haussler et al for the binary case. For proper losses, this was subsequently generalized via the Hessian of the Bayes loss by van Erven et al to the multi class case.

This work provides (convex) geometric characterization of the notion of mixability for proper loss functions. Specifically, a central result here is that a loss is mixable iff its (argument-scaled) super prediction set slides freely inside that of the log loss. This distinctiveness of the log loss in particular ties with a previous result of Vovk establishing that log loss is the “most” mixable.


**Audience:**

Yes

**Broader Impact Concerns:**

None.

**Claims And Evidence:**

Yes

**Requested Changes:**

The paper is well-written, but I found it to assume too much on the sequential prediction side (ironically), and market itself a bit narrowly. For example, it might be pedagogically better to introduce the definition of mixability using substitution/witness functions (like in http://onlineprediction.net/index.html?n=Main.Mixable or in Definition 6 of Mhammedi et al, even as the current definition is equivalent); that would make its applicability to SAA more transparent.

**Strengths And Weaknesses:**

Strengths
+ The characterizations here for proper losses is conceptually “right” in that it is paramertization-independent.
+ A demonstration of the strength of this interpretation lies in that it extends seamlessly (i.e., through sensible generalizations) to the multi class case; the same can’t be said about all other equivalent conditions we know.

Not-Strengths
- Beyond stating the result, it would be great to demonstrate the utility of these characterizations by using them to (re)establish the mixability eta’s for specific loses.
- This point is not a “con” in any strict sense. The contribution here is in establishing the correct interpretation, and is not algorithmic. However, I write this in full awareness of the fact that an improved understanding that paves way for further research is equally (if not more) valuable.

Overall, although the breath of contributions here is narrow, I think the result will appeal to the sub community (online learning) in question, and I hope that eventually members will further build on it.

---

### Review · Reviewer_R4xj · 2023-06-12

**Summary Of Contributions:**

The paper presents a geometric characterisation of mixable games.

**Audience:**

Yes

**Broader Impact Concerns:**

None.

**Claims And Evidence:**

Yes

**Requested Changes:**

As the main result of the paper is certainly correct, I have no problem with the publication.

**Strengths And Weaknesses:**

The main result of the paper is certainly correct but the paper takes a convoluted way to arrive at it.

The statement is easy to prove in much higher generality. Mixability is equivalent to convexity of the image of the set of superpredictions under the exponential transformation $x_i \to e^{-\eta x_i}$. As the set of superpredictions has non-empty interior, convexity is equivalent to the existence of support hyperplanes: through every point on the boundary there passes a support hyperplane, i.e, a hyperplane such that the the set is entirely on one side of it. The inverse image of this support hyperplane is the boundary of the shifted set of superpredictions for the logarithmic game.

This simple argument makes no reference to smoothness of the loss function. It dispenses with the requirement of the propriety of the loss either.

The real problem here is to find a good characterisation of the mixable losses in terms of the geometry of the original space. The excellent paper by Van Erven, Reid, and Williamson makes a very good point in terms of the curvature.

---

### Review · Reviewer_cZM7 · 2023-06-19

**Summary Of Contributions:**

The goal of this paper is to provide interpretations for mixable loss functions in the context of prediction with expert advice by differential geometry

**Audience:**

Yes

**Broader Impact Concerns:**

No concerns that I am aware of.


**Claims And Evidence:**

No

**Requested Changes:**

Please refer to the weakness part for changes. Moreover, there are some minor changes including typos, unclear notations, repeated definitions using the same symbols, etc.

**Strengths And Weaknesses:**

Strengths
1. The main theoretical results are theoretically sound and correct. Did not find major flaws in the proof (except for some notational issues).
2. Theorem 4.22 and the equivalent conditions for \eta-mixable loss functions and the geometric properties of loss functions are interesting.
3. The reconciliation between this work and previous works in Section 1.6 is interesting, providing a new point of view for mixing games.

Weaknesses
1. Motivation is unclear to me: The authors state that the motivation came from the observation made by [Vovk2015]. However, the intuitive explanation is missing here for why the observation could have a broader impact on mixing games. How could this new geometric interpretation potentially help us better understand or design games?
2. The presentation is not ideal: The authors introduce mathematical definitions at the very beginning of the paper, which I think is good and clear, but then take a long detour in terms of arriving at the main contribution and the proposed differential geometry interpretation. More direct language is recommended, for example, the smoothness (i.e., the curvature) of the loss function is a sufficient condition for the mixability of games. The complicated content an layout of this paper makes it hard to follow.
3. The benefit of this new viewpoint is not strong enough. Proposing a new viewpoint that we are unclear about before, in this case, the geometry, is always welcome. However, this new interpretation should not be used just as a viewpoint, but as a practical tool that provides us with new guidelines, e.g., designing \eta, finding the limiting number of games that could be mixed, or converse results such as inmixability.
4. Future direction is not concrete: it is highly recommended that the authors point out potential directions to follow up this paper so that to maximize its impact.
5. As an online learning paper, I also think it is reasonable to ask for a few illustrative experiments for a proof of concept for the theory.  Readers could potentially learn more from it.

---

### Decision · Action_Editors · 2023-08-08

**Recommendation:** Accept as is

**Comment:**

The reviewers all recommended acceptance. The paper is quite technical, but it is dealing with a concept -- mixability -- that is fundamental to the subject of sequential decision making.

The reviewers raised issues around motivation. The authors have generally stuck to defending their current approach to motivation (offering a coordinate free way of reasoning / understanding mixability), which I believe is fine and justified. The authors don't offer any immediate computational consequences, but this conceptual advance should pave the way for others to deliver algorithmic ones.

**Audience:**

The audience for the paper is theoreticians working in machine learning, and especially those working within sequential prediction, although the notion of mixability is more generally applicable.

**Claims And Evidence:**

The claims are theoretical and they are proven rigorously.